# On the Interpretation of Synthetic Aperture Radar Images of Oceanic Phenomena: Past and Present

**Kazuo Ouchi [1,*] and Takero Yoshida [1,2]**

[1] Former Institute of Industrial Science, The University of Tokyo, Kashiwa-shi, Chiba 277-8574, Japan
[2] Department of Ocean Sciences, Tokyo University of Marine Science and Technology, Minato-ku, Tokyo 108-8477, Japan
* Correspondence: ouchi.kazuo@gmail.com

**Abstract:** In 1978, the SEASAT satellite was launched, carrying the first civilian synthetic aperture radar (SAR). The mission was the monitoring of ocean: application to land was also studied. Despite its short operational time of 105 days, SEASAT-SAR provided a wealth of information on land and sea, and initiated many spaceborne SAR programs using not only the image intensity data, but also new technologies of interferometric SAR (InSAR) and polarimetric SAR (PolSAR). In recent years, artificial intelligence (AI), such as deep learning, has also attracted much attention. In the present article, a review is given on the imaging processes and analyses of oceanic data using SAR, InSAR, PolSAR data and AI. The selected oceanic phenomena described here include ocean waves, internal waves, oil slicks, currents, bathymetry, ship detection and classification, wind, aquaculture, and sea ice.

**Keywords:** synthetic aperture radar (SAR); polarimetry; along-track interferometry; oceanic phenomena; ship detection; aquaculture

## 1. Introduction

The ocean covers the two thirds of the surface of the earth, having a strong impact on the global climate through carbon absorption and heat transfer. The ocean is also an important source of food and maritime transport. Since the first civilian spaceborne synthetic aperture radar (SAR) onboard the SEASAT satellite in 1978 with its main purpose of ocean monitoring, a substantial number of spaceborne and airborne SARs have been launched and used in practice over ocean as well as land [1–4]. For the ocean application, an excellent summary was presented by Jackson and Apel in 2004 [5], covering the principal theory of SAR and applications to most of the oceanic phenomena. New technologies have also been developed, including cross-track interferometric SAR (InSAR) for production of a digital elevation model (DEM) of land surface and surface deformation measurements [4,6–10], along-track InSAR (ATI SAR) applied to the measurements of the velocity of moving targets [11–16], polarimetric SAR (PolSAR), and polarimetric interferometric SAR (PolInSAR), mainly for image classification [17–22], as well as the recent applications by artificial intelligence (AI) such as deep learning [23–29].

An example is shown in Figure 1 of the ERS-1 C-band VV-polarization SAR image of the English Channel and Isle of Wight. The image shows several oceanic features such as ocean waves, warm and cold water boundary, areas of different ocean wind speeds, the effect of underwater topography, ships, and a dark linear feature of possible oil spills dumped by a ship. These and other oceanic phenomena can be detected and their properties can be estimated by the intensity data associated with the spatial and temporal changes of sea surface.

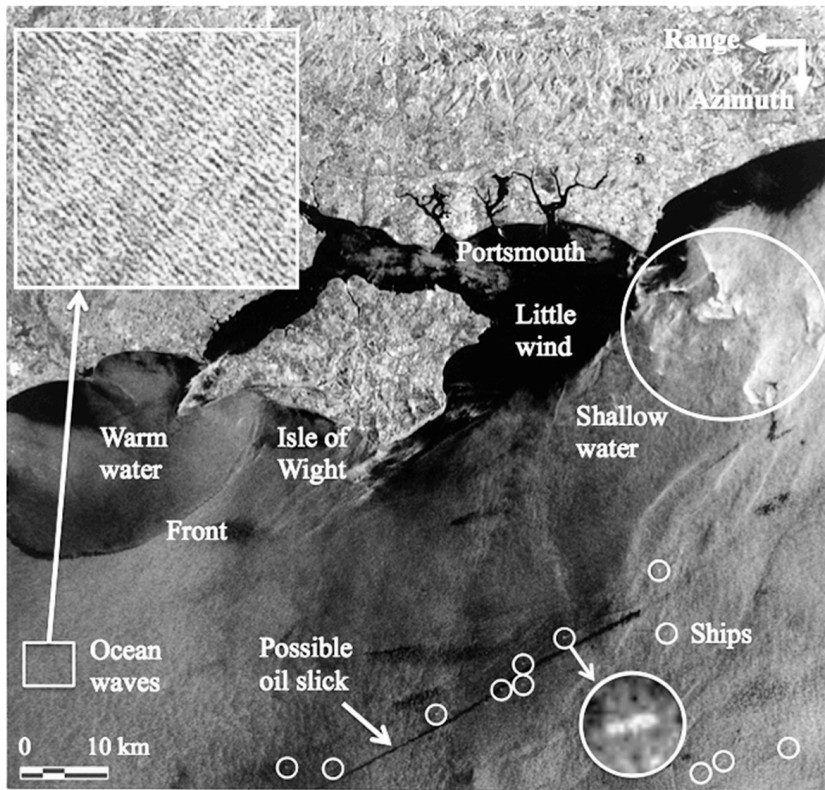

**Figure 1.** ERS-1 SAR image of the English Channel (scene center: 50.57°N, 1.17°W), showing several oceanic features. (Courtesy of ESA).

The complex SAR data can be used for the velocity measurement of ocean currents through the Doppler centroid shift of the azimuth spectra. Another popular method is the ATI SAR, with two antennas placed in the along-track direction on aircraft or spaceborne SARs in tandem, such as TerraSAR-X and TanDEM-X [12,13]. These methods can measure only the range velocity component of ocean currents, while a recent proposed method based on multi-look processing of ATI SAR is able to estimate the velocity vector. This multi-aperture ATI SAR (MA-ATI SAR) produces the forward- and backward-looking interferograms using multi-look processing of two ATI SAR data, yielding the velocity components in the corresponding directions, and thus the velocity vector of ocean currents [15,16]. Another method of velocity vector measurement was proposed using two antennas on a single platform, where the forward and backward beams are transmitted by a main antenna, and the backscattered signal is received by the two antennas. By using the Ku-band of short wavelength, an antenna separation of 12 m is possible on a spaceborne platform [14].

PolSAR and PolInSAR can be applied mainly to land-cover image classification through different backscattering processes depending on different polarization combinations. For the ocean application, PolSAR is used for detection and classification of manmade targets such as ships, marine cultivation, and also sea ice and oil slicks [30–39].

Further, in recent years, AI has attracted much attention for several oceanic applications. The study of AI such as Perceptron started in 1950s, followed by machine learning in the 1980s to the early 2000s [40]. The recent popular algorithms of deep learning include convolution neural network (CNN) composed of multi-layer neural networks (NN) [26]. In general, these algorithms require a large quantity of training data. For ocean applications, the deep learning method is often applied to detection and classification of internal waves, oil spills, ships, and sea ice, as well as the estimation of waveheight, e.g., [41–45].

In the present article, a brief description is first given on the principle of radar backscatter from the sea surface, which is the basis of many applications. A summary is then given

on the applications of the conventional SAR data, ATI SAR, PolSAR data, and artificial intelligence over oceans, including surface waves as well as breaking waves, oceanic internal waves, ocean currents, aquaculture, and ice. It should be noted that there have been substantial numbers of publications to date on the SAR/Ocean. In the present review, only a limited numbers of references are given, and for further readings the references therein are recommended.

## 2. Ocean Surface Waves

### 2.1. Mirowave Backscatter from Sea Surface

In order to describe the imaging process of ocean waves, it is useful, first, to explain briefly the radar backscatter from the sea surface. Since the penetration depth of the microwave used by SAR is a few millimeters for the seawater, the SAR images contain only the information on the surface of the sea. As noted in the preceding section, the intensity data are proportional to the microwave backscatter from the sea surface, and the power of backscatter is related to the spatial and temporal variations of oceanic phenomena.

The elementary scatterers which give rise to the radar backscatter from the sea surface are the small-scale waves satisfying the Bragg resonance condition $\lambda_{Bragg} = \lambda/(2\sin\theta_i)$, where $\lambda$ is the radar wavelength and $\theta_i$ is the incidence angle. For example, for X-band of $\lambda = 0.03$ cm and L-band of $\lambda = 0.23$ cm with $\theta_i = 35°$, the Bragg wavelengths are 2.6 cm and 20 cm, respectively. The Bragg resonant radar backscatter was first observed by D. D. Crombie by a ground-based HF radar in 1955 [46]. Since then, there have been extensive theoretical and experimental studies on the radar backscatter from sea surface under different environmental conditions in terms of radar incidence angles and polarizations, e.g., [47]. Temporal and spatial changes of the Bragg waves associated with, for example, local surface tilt angles of ocean waves and damping of small-scale waves by oil slicks, give rise to the varying radar backscatter and hence image modulations.

### 2.2. Imaging Process of Ocean Surface Waves

It is interesting to note that prior to the launch of SEASAT in 1978, a question was raised whether the spaceborne SAR could produce fine images of moving ocean waves on a global scale. However, as is known, the SEASAT-SAR proved its ability to image dynamic ocean waves, and considerable efforts were made on the theoretical and experimental studies during the late 1970s to the early 2000s. Consequently, the principal theory of imaging ocean waves by SAR is well understood with supporting experimental data, followed by several spaceborne SARs as routine wave monitoring [48–56].

Figure 2 is a SEASAT-SAR image of the Fair Isle in the North Sea and the surrounding water, showing ocean waves propagating from left to right in the range direction and those refracted around the island. Four imaging processes are known to interpret these wave images, including the tilt, hydrodynamic, foreshortening and velocity-bunching modulations.

As in Figure 3, the tilt modulation is due to the changes of the local incidence angle on the surface of ocean waves, in which the backscatter radar cross section (RCS) is large from the wave surface facing toward the radar, and it is small from the surface facing backward. Therefore, this modulation is largest for range-travelling waves, and almost null for azimuth-travelling waves. The hydrodynamic modulation is caused by the water movement, where the surface water on the wavefront facing toward the wave propagation direction converges, giving rise to a rough surface, while the surface facing backward becomes smooth. The image modulation is weak in comparison with the tilt modulation, as illustrated in Figure 4. The foreshortening modulation is, as the name suggests, due to foreshortening of wave undulation. Again, this modulation is weak and characteristic of the range-travelling waves.

The very strong modulation by velocity bunching is caused by the azimuth image shift of the moving scatterers having different slant-range velocity components. This highly non-linear modulation is dominant for azimuth-travelling waves and is absent in range-

traveling waves, as in Figure 4a. Velocity bunching is well known where the image of a scatterer having the slant-range velocity $v_{SR}$ [m/s] is shifted in the azimuth direction by the distance $\delta X = -(R/V)v_{SR}$. Here, R and V are the slant-range distance from the antenna to the scatterer and platform velocity, respectively. For example, for $R = 750$ km, $V = 7$ km/s, the amount of shift is $\delta X = -107\ v_{SR}$ [m].

Figure 4b illustrates the velocity bunching modulation. As the wave propagates the scatterer at a position A moves downward between the time $t_1$ and $t_2$, while the scatterer B moves upward, i.e., the Bragg waves have negative and positive slant-range velocity components depending on the positions and also the wavelength, amplitude, and phase speed of ocean waves. Then, the images of the scatterers are concentrated or dispersed in the image plane, yielding wave-like images although the RCS is uniform. Depending on the amount of shift, double peaks may also appear. Experimentally, the effect of velocity bunching was proven by the waves over floating sea ice [50].

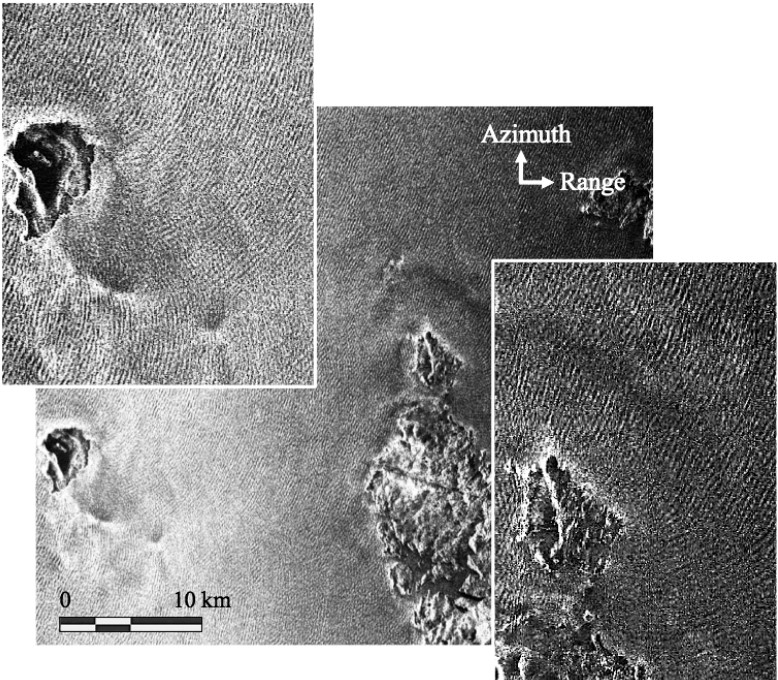

**Figure 2.** SEASAT-SAR image of the North Sea in northern Scotland, where the island of Foula is on the left and the mainland on the right. The images of ocean waves can be seen propagating from left to right in the range direction, and those refracted around the islands in the enlarged images.

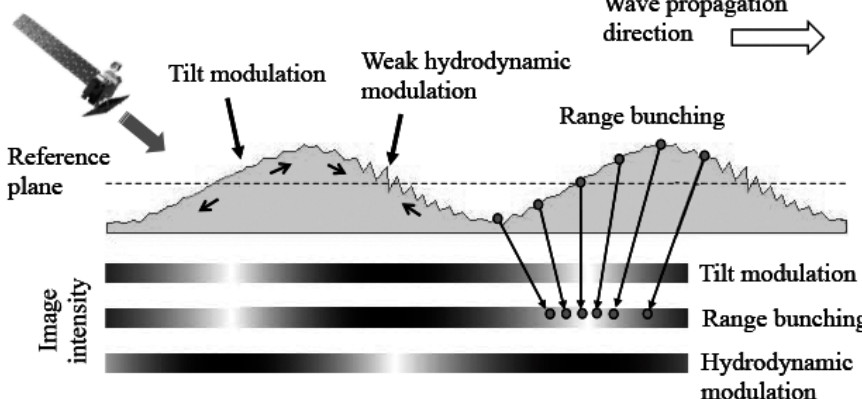

**Figure 3.** Illustration of the imaging processes of range-traveling ocean waves by the tilt, weak hydrodynamic and foreshortening modulations.

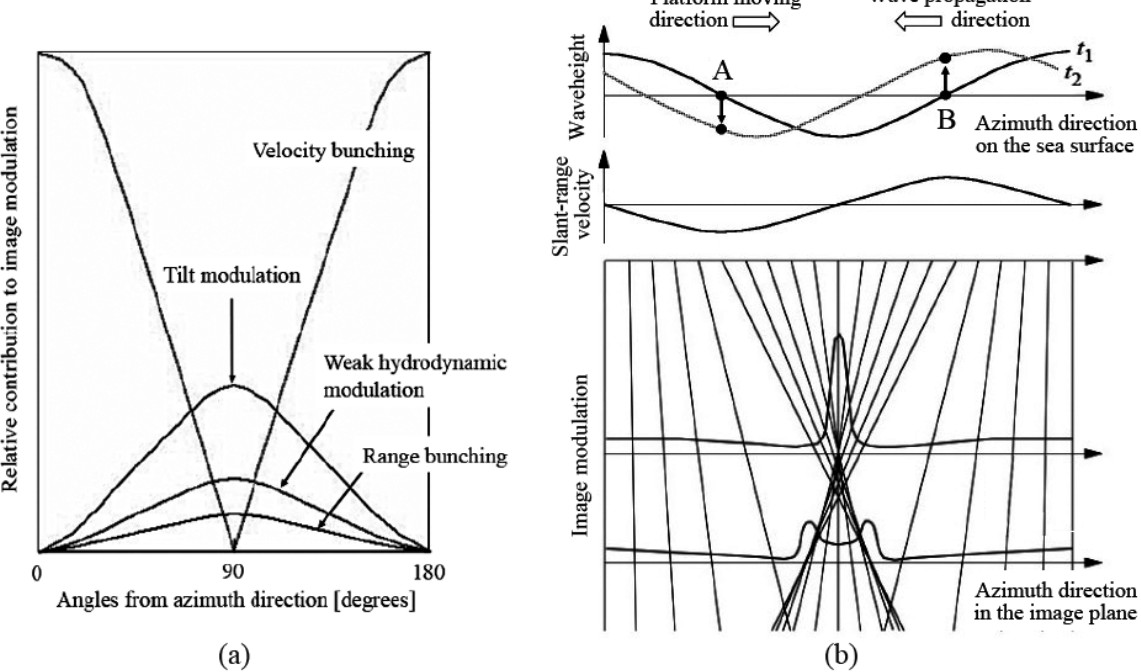

**Figure 4.** (**a**) Relative contribution to ocean wave image modulation in terms of the azimuth angle. (**b**) Illustration of the non-linear image modulation by velocity bunching.

### 2.3. Directional Wave Spectrum

Despite the non-linearity of ocean wave imagery, the measurement of the directional wave spectrum based on multi/sub-look processing has been proposed and used in practice [53–56]. The initial purpose of multi/sub-look processing is to increase the signal-to-noise ratio (SNR) by adding sub-look images on an intensity basis. Since the inter-look speckle patterns are not correlated, the standard deviation of speckle noise is reduced by $1/N^{1/2}$ at the expense of the reduction in image resolution by $1/N$. In this way, the images of stationary targets can be enhanced, but the images of moving targets such as ocean waves cannot be improved as much as for the stationary targets because of different inter-look image positions.

To produce the directional wave spectrum from the two-look processing, for example, independent sub-look reference signals are applied to the raw data yielding two sub-look images, as illustrated in Figure 5. The corresponding spectra $S_1$ and $S_2$ are computed by Fourier transform (FT). Then, $S_1 \times S_2{}^*$ is multiplied by a weighting function of the phase proportional to the inter-look time difference $\delta t$ and a preset radian frequency of ocean waves, where the asterisk * implies taking a complex conjugate. The resultant weighted wave spectrum removes the directional ambiguity from the conventional wave spectrum produced by Fourier transforming the full-look image [53–56]. Here, an example is given using two-look images, but several sub-look images can be used to increase the measurement accuracy using SAR data, such as those of Spotlight SAR of long azimuth integration times. In the sub-look processing described above and in Figure 5, sub-images are produced from the raw SAR data.

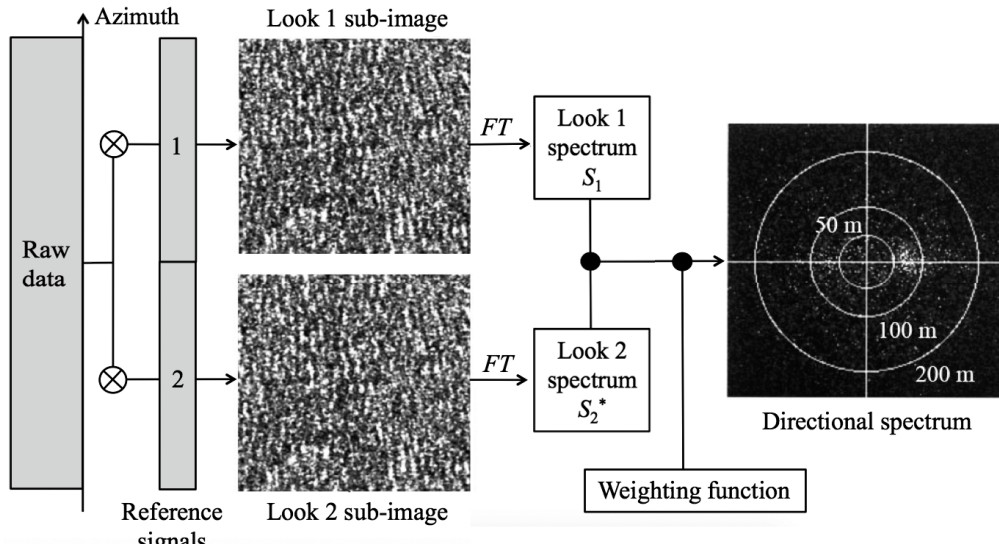

**Figure 5.** Measurement of directional wave spectrum using the method of weighted cross-spectra.

The directional wave information can also be obtained by taking the cross-correlation of sub-images. The result yields not only the wave propagation direction but also wavelength [53].

Another method of sub-look processing is to divide the full-look complex image spectrum into the multiple sub-spectra and by inverse Fourier transform of the individual sub-spectrum. A further detailed process will be given in Section 7 (Ship Detection).

### 2.4. Waveheight

Apart from the directional wave spectrum, the significant waveheight is an important parameter. There have been several attempts to estimate waveheight using the images of waves in ice, RCS of range-travelling waves, and HH/VV-polarization ratio, but they are not proven for practical applications. However, a recent study reported using a deep neural network (DNN) for waveheight estimation [57]. In the study, a dataset of over 750,000 collocations between radar altimetry and the two Sentinel-1 SAR sensors was created, and used to train a DNN regression model. By training with the wave cross-spectra from the collocation data acquired from 2015 to 2017, the model was demonstrated on test data acquired during 2018. This deep learning model reduced the mean squared error by 50%, from 0.6 m to 0.3 m, when compared with the altimeter data, and the root mean square error (RMSE) of the significant waveheight was 0.3 m as compared with the independent altimeter observations. It was noted that there is still room for improvement with additional training data in the extreme sea states with significant waveheights larger than 8 m.

The significant waveheight estimation by CNN was reported based on the relation between the normalized RCS and significant waveheight [58]. After training, the comparison of the test results using 1597 C-band Sentinel 1 SAR images with the matched in situ buoy data indicated a mean RMSE of 0.32 m with a correlation coefficient of 0.90.

The AI algorithms such as CNN are also used in other ocean applications. Further details on AI are given in Section 4.3.4.

### 2.5. Breaking Waves

The ocean waves described so far are the wind waves generated by the energy input to the water surface by wind. With increasing wind speed, duration and fetch, the waves around the crest start to break, containing air bubbles. These breaking waves can also be seen in shoaling waves and waves broken by rocks. When breaking waves are imaged by SAR, they appear as bright linear images in the azimuth direction, known as "azimuth streaks".

Figure 6 is the RADARSAT-2 X-band SAR image of Chuuk Lagoon, Micronesia, acquired on the 4th of November, 2010. Ocean waves propagating from the right to left are diffracted by the inlet toward inside the lagoon. The bright linear images along the coastal barrier are the azimuth streaks caused by braking waves.

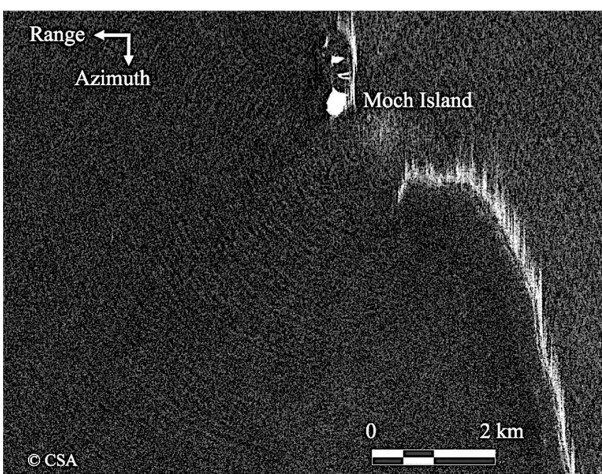

**Figure 6.** RADARSAT-2 X-band SAR image of Chuuk Lagoon, Micronesia, showing the bright azimuth streaks caused by breaking waves along the coastal barrier.

Figure 7 illustrates the origin of azimuth streaks. The top-left image shows the azimuth Doppler phases from a stationary point scatterer and a scatterer in random motion. The corresponding point images (PSFs: point spread functions) are in the bottom-left image. Because of the random motion of the point scatterer, the PSF is smeared in the azimuth direction due to the azimuth image shift, explained in Section 3.1, for velocity bunching of surface waves. Because of different range velocities of the point scatterer, the smeared PSF contains random fluctuation.

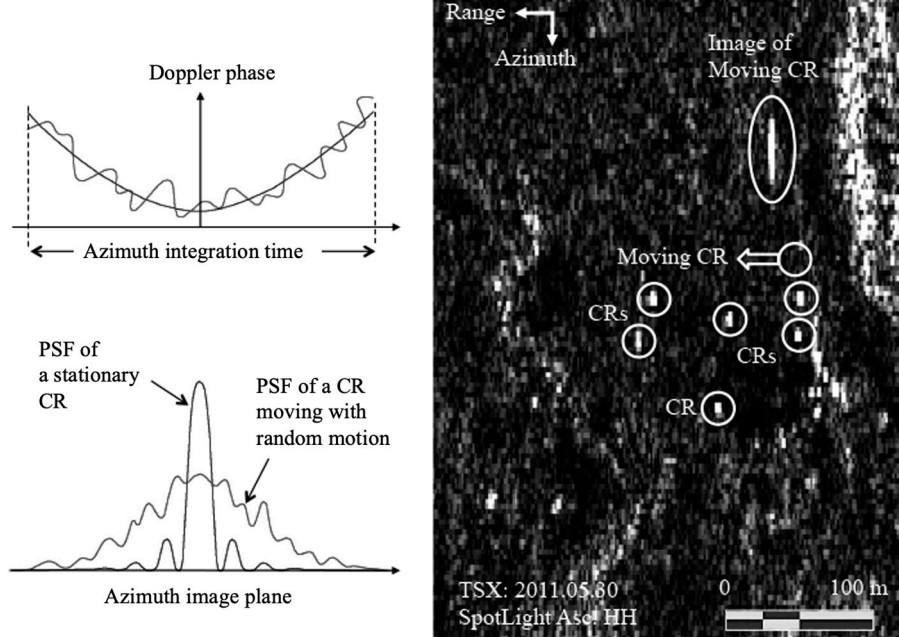

**Figure 7. Top left**: Azimuth phases of a stationary point scatterer (CR: corner reflector) and the point scatterer moving in random motion. **Bottom left**: The corresponding point spread functions (PSF) of the stationary and moving CRs. **Right**: TerraSAR-X image showing stationary and moving corner reflectors.

The right image in Figure 7 is the TerraSAR-X SpotLight image of the baseball ground of the National Defense Academy, Japan. During the observation, several corner reflectors (CRs) were placed, and a CR moved in the range direction with varying speed. The image of the moving CR is shifted by the distance proportional to the mean velocity, and spread in the azimuth direction due to the random velocity changes. This is the basis of the azimuth streaks caused by breaking waves.

## 3. Oceanic Internal Waves

### 3.1. Observation and Impact

The oceanic nonlinear (non-sinusoidal) internal waves (IWs), known as "dead water" were first recorded by the Norwegian explorer F. Nansen in 1893, and later explained by Ekman by laboratory experiments in 1904 [59,60]. IWs are generated at the interface of stratified water layers (i.e., pycnocline) often induced by the interaction between currents and bottom topography. The stratification can be caused by high- and low-density layers associated with differences in temperature and/or salinity [61–64].

Observations of nonlinear IWs by optical sensors have been reported since the 1950s, and a large number of studies using both optical and SAR have been reported since the launch of SEASAT in 1978. The IWs appear as a single solitary wave also known as a soliton or a packet of multiple nonlinear internal waves. The largest waves in wavelength and amplitude appear at the front of the wave packet, decreasing toward the trailing edge, and propagate over several hundred kilometers with phase velocities ranging from 0.1 m/s to several m/s.

Figure 8 shows the ENVISAT-ASAR images (left column) and MODIS-Terra images (right column) of two IW packets in the two areas (A and B) of the South China Sea. The MODIS images were acquired at 55 min after the ENVISAT-ASAR data. These IWs were generated by the interaction of tidal currents with the edge of continental shelf at the Luzon Strait between Taiwan and Luzon island of the Philippines, propagating from right to left toward the east coast of China. The two wave packets are separated by the distance proportional to the semi-diurnal tidal period of 12.42 h. As will be discussed later, the phase velocities of these IWs can be estimated from the distance between the two wave packets and tidal period.

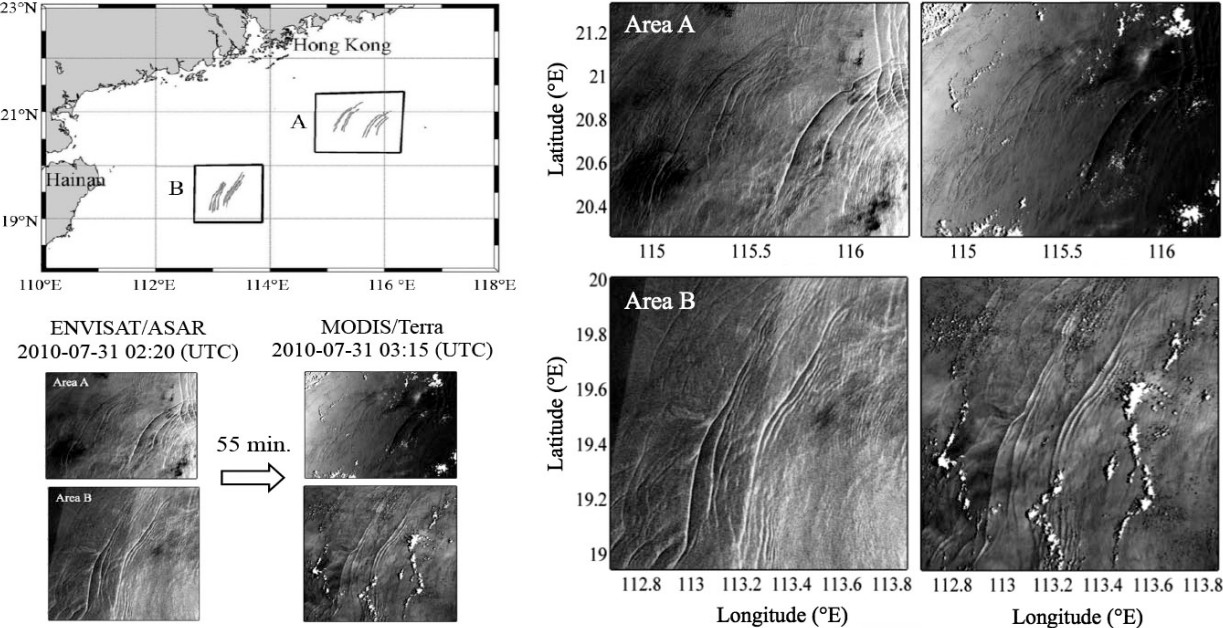

**Figure 8.** ENVISAT-ASAR and MODIS-Terra images of the South China Sea acquired on 31 July 2010 with 55 min time difference. Two tidally generated internal wave packets are separated by the distance proportional to the semi-diurnal tidal period [64].



Substantial numbers of in situ surface and in-water measurements have also been made by several instruments such as echo-sounder profilers, acoustic Doppler current profilers (ADCP) and conductivity temperature depth profilers (CTD). Some of these measurements were used to validate the simultaneous SAR observations, e.g., [65–68].

Oceanic internal waves are of great interest in the field of geoscience for their strong energy transport between the deep water and continental shelf, nutrient mixing, sediment re-suspension, effects on marine architecture, underwater communication, and also on global climate [69–73]. For example, the dissipation energy of the semi-diurnal tides (M2) is approximately 2.4 TW, which is 2/3 of the total energy, and the semi-diurnal tides generate IWs of 0.3–1.2 TW energy. The IWs generated in the Lombok Strait are considered to have substantial effects on the EL Niño and EL Niño Southern Oscillation (ENSO).

The mechanism is based on the strong seasonal energy transport by IWs between the Java Sea and Indian Ocean through the Lombok Strait, where the both monsoon- and El Niño-induced variability is strong. As such, the throughflow likely varies on interannual time scales to reflect an ENSO influence by reduced and enhanced transports during EL Niño years [70].

### 3.2. Imaging Process

Since the microwave does not penetrate into the interior of the sea, SAR does not directly measure IWs. Figure 9 illustrates the imaging process, where the left and right figures correspond, respectively, to the nonlinear IWs of depression and elevation. As the IW propagates from left to right, water particles follow a circular motion, and this motion induces varying surface currents. The surface of converging currents becomes rough, and the radar backscatter is large compared with the smooth surface of diverging currents. Thus, the SAR images are manifestation of IWs.

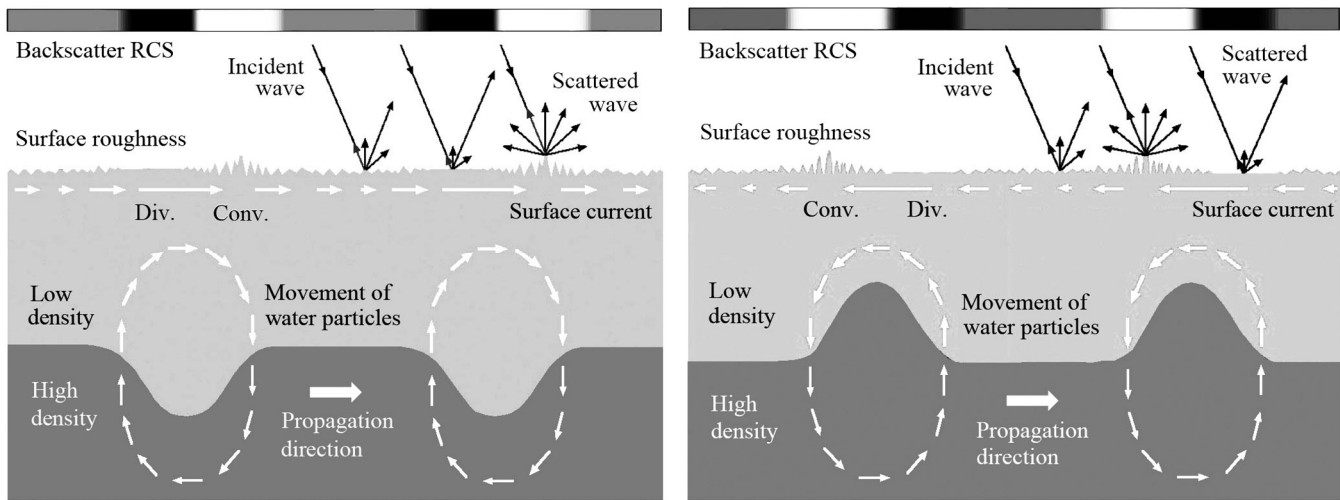

**Figure 9.** Imaging process of IWs by SAR. The varying surface roughness is caused by the circular motion of water particles as the internal wave propagates, resulting in the current converging and diverging areas. The current converging surface becomes rough compared with the diverging area. The image signatures depend on the depression (**left**) and elevation (**right**) of the pycnocline.

The SAR image signature of nonlinear IWs depends on the depression and elevation of the pycnocline, e.g., [61,62]. The most common pattern is the images of IWs of depression, shown in the left of Figure 9, where a bright band is followed by a dark band in the direction of propagation on the gray background, as in Figure 8 and the top-left image of Figure 10. For the nonlinear IWs of elevation, the bright band appears behind the dark band in the direction of propagation, as illustrated in the right of Figure 9 and the ERS-1 SAR image in Figure 10. This reversal of bright and dark images may occur, such as the case where internal wave packets propagate toward shore on the continental shelf of decreasing

water depth. As internal waves approach shallow waters, the leading undulation may decay, and the pycnocline tends to become uniform, resulting in the dark band leading the bright band [62,63,65].

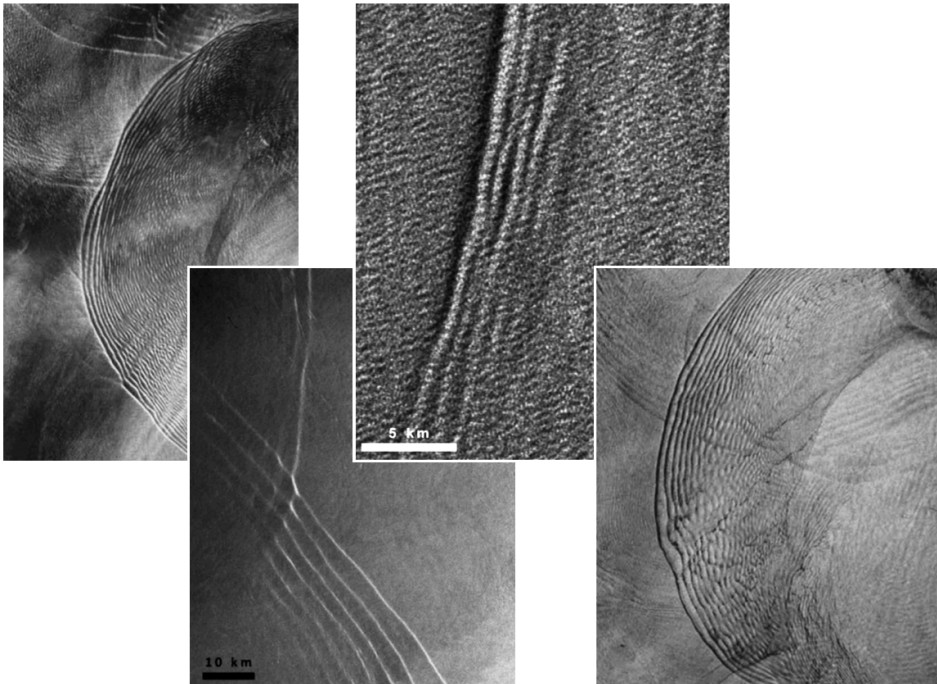

**Figure 10.** Different image signatures of nonlinear internal waves [63]. **Top left** and **bottom right**: TerrasAR-X images over the Gulf of Main west of Cape Cod acquired, respectively, on the 23rd of June and 3rd of July, both in 2008. **Top right**: ERS-1 SAR image over the Mozambique Channel acquired on the 24th of September 2001. **Bottom left**: ENVISAT-ASAR image over the Andaman Sea acquired on the 18th of November 2006. The propagation direction is from right to left, except the ENVISAT-ASAR image, in which the interaction can also be seen between the two wave packets propagating predominantly from left to right.

Another mechanism associated with ocean wind and surface films is biogenic oil [62,63,73,74]. The imaging processes shown in Figure 9 are under the condition of clean sea surface under moderate wind speed. If the biogenic oil is present on the varying currents induced by IWs, the oil is concentrated in the current convergence zone. Since oil slicks have higher viscosity than seawater, they damp the small-scale Bragg waves.

If the power of damping of surface roughness caused by the current convergence is large in comparison with the background surface, the SAR image shows only a dark band on the gray background as in the bottom-right image of Figure 10.

The wind speed also affects the image signature of IWs. Under the low wind condition, the surface becomes moderately rough on both the current converging and background surfaces with and without surface films, yielding only bright bands on the gray background, as in the ENVISAT-ASAR image in Figure 10.

As to the detection of IWs in SAR images, it is interesting to note that faster regions-CNN (R-CNN) [24] was used for IW detection utilizing the 466 ENVISAT-ASAR images over the South China Sea acquired during 2003 to 2012. A total of 888 IW images were trained to identify IWs, resulting in a 94.78% recognition rate. The computation time for detection was 0.22 s/image, suggesting the capacity of CNN for the automatic detection of IWs [75]. Further details on AI are given in Section 4.3.4.

*3.3. Estimation of Internal Wave Properties*

As mentioned with reference to Figure 8, the phase velocities of IWs can be estimated from either the multiple IW images separated by the semi-diurnal tidal period in a single

SAR/optical data, or the measurement can also be made from two sets of data separated by a short acquisition time [64,76]. Figure 11 shows the results of estimating the phase velocities of IWs in Figure 8, where the measurements were made using (a) the difference between the ENVISAT and MODIS images with a time difference of 55 min, and (b) those using the IW packets with a time difference of 12.42 h within the ENVISAT and MODIS images. Comparison is made with the theory first proposed by Munk [77], which describes the depth-dependent IW propagation with the two-layer vertical structure function of an IW. Numerical computation was made to produce the theoretical results in Figure 8 using the water temperature, density, and depth [64]. Agreement with the theory is good for (a), with the correlation coefficient of 0.81, while (b) shows a moderate correlation of 0.59. It is clear that the good correlation of (a) is due to the short time difference between the ENVISAT and MODIS image acquisition times. Note that the two wave packets are separated by a longer distance in area A than area B in Figure 8. The reason may be associated with different tidal powers, and/or IWs in area B being further from the Luzon Strait, travelling over longer distance of varying water depths.

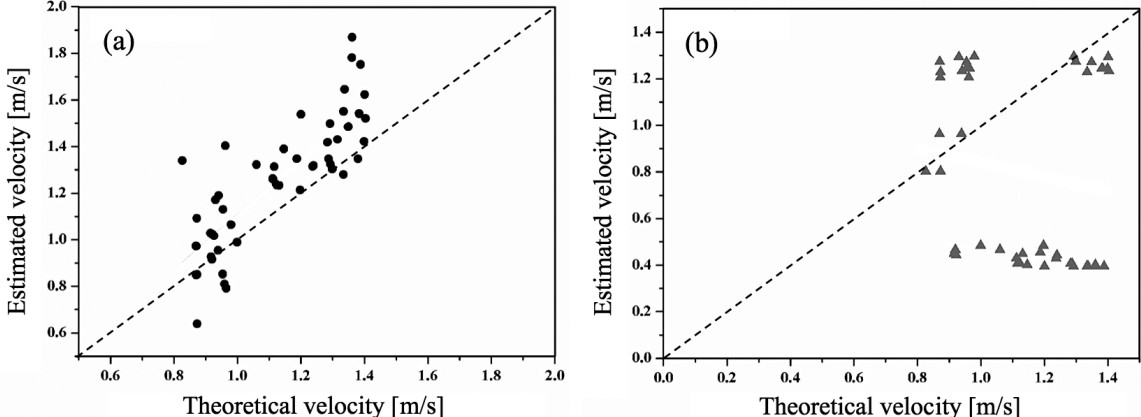

**Figure 11.** Estimated and theoretical IW phase velocities of selected points in Figure 8. (**a**) Velocities estimated using the difference between the ENVISAT and MODIS images with a time difference of 55 min. (**b**) Velocities estimated using the IW packets with a time difference 12.42 h [64].

Similar results have been reported by several authors. For example, Liu, et al. [76] used the Envisat and ERS-2 tandem data separated within 30 min in the South China Sea of water depths ranging from 100 m to 4000 m. The estimated phase velocities agreed well with the theoretical values, and they were affected mainly by bottom topography. The relation of IW velocity and water depth was also investigated, with good agreement with the theory.

Figure 12 is an example of the theoretical interpretation and extraction of the properties of IWs from a single SAR image. The initial values are of a soliton amplitude, upper and lower layer densities, and the water depth. Then, the Korteweg-de Vries (K-dV) equation [77–79] is applied. The K-dV equation is a non-linear partial differential equation, and provides analytical solutions under certain conditions. The solutions describe the interface displacement of a soliton in terms of the phase velocity and dispersion coefficients between the two layers in the water of a constant depth. A modified K-dV equation can take into account the effect of water heterogeneity and depth, which is also used for simulation study [80–82].

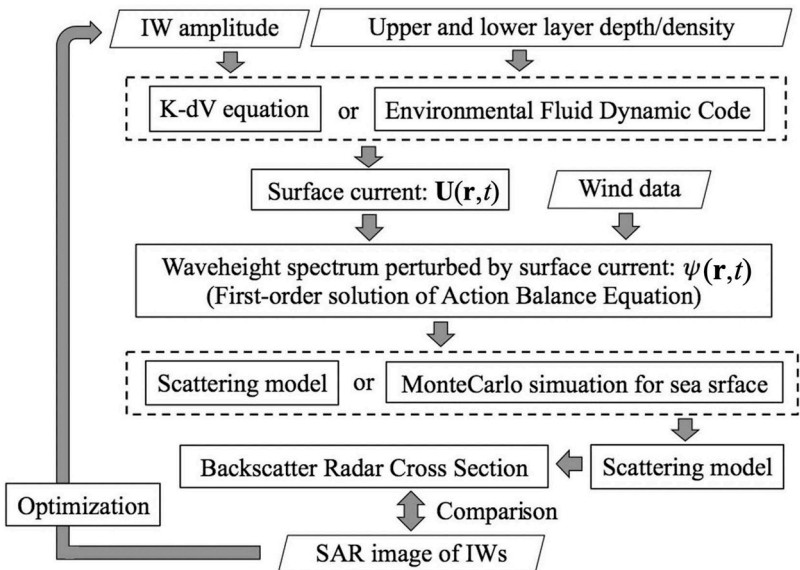

**Figure 12.** An example for the analysis of SAR images of IWs.

Using the K-dV equation, the surface current vector $\mathbf{U}(\mathbf{r},t)$ can be computed where $\mathbf{r}$ and $t$ are the spatial vector and time variable, respectively. Another approach is using the environmental fluid dynamic code (EFDC) that simulates the three-dimensional flow process in surface water systems in a finite difference computational grid.

Having computed the spatial and temporal changes of surface currents induced by the IWs, the relaxation time approximation, which is the first-order solution of the action balance equation, is used with wind data to describe the waveheight spectrum in terms of the varying surface currents. This approximation describes the evolution of small amplitude of the sea surface by a slowly varying current. The Monte-Caro method can also be applied to simulate the spatial and temporal changes of the perturbed sea surface. Radar backscatter models are then applied to the sea surface using the waveheight spectrum perturbed by the IW. Scattering models such as the physical optics, small perturbation, and integral equation methods can be used to compute the backscatter RCS. The results are compared with the SAR data, and optimization, if necessary, is made by iteration. This approach is generally used to interpret the SAR images and extract the properties of IWs, e.g., [76,81], including the theoretical velocity shown in Figure 11.

It is worth mentioning at this stage about the early SAR Signature Experiment (SAR-SEX) conducted in the New York Bight in 1984, where multi-frequency airborne SAR and in situ data of IWs were collected. Using the Bragg scattering model, good agreement was obtained between the theory and observed intensity modulation with L-band data, but the X-band intensity modulation was underestimated by an order of magnitude in comparison with the observed intensity modulation. The reason for the discrepancy was considered to be the additional small-scale roughness perturbed by meter-scale waves by the current field induced by IWs [83], while good agreement was obtained both at X- and L-band using the same SAESEX data with the physical optics scattering model [84]. The reason is that the Bragg model is based only on the small-scale waves, while the physical optics model takes into account the waves of all wavelengths as well as the Bragg waves.

A further interesting result on the frequency-dependent SAR images was obtained during the joint UK-USA controlled experiment for the validation of ship-generated IWs in Loch Linnhe, Scotland, during 1989–1994 [85]. Airborne SAR data of six frequencies ranging from Ka-band (~35 GHz) to P-band (~0.5 GHz) were collected with simultaneous in situ measurements. Different scattering models were also considered, including the modified Bragg, Kirchhoff (physical optics), and composite surface models [85–87]. The upper image of Figure 13 shows the X-band SAR image of IWs and surface and in-water instrumented sites. The lower figure is the relative image intensity modulation in

P-, L- and C-bands. It can be seen that the P-band image is at an advanced position followed by the C-band and X-band images. This is another proof that the principal scatterers are the Bragg waves, i.e., the P-band Bragg waves have longer wavelength and propagate faster than the others [87].

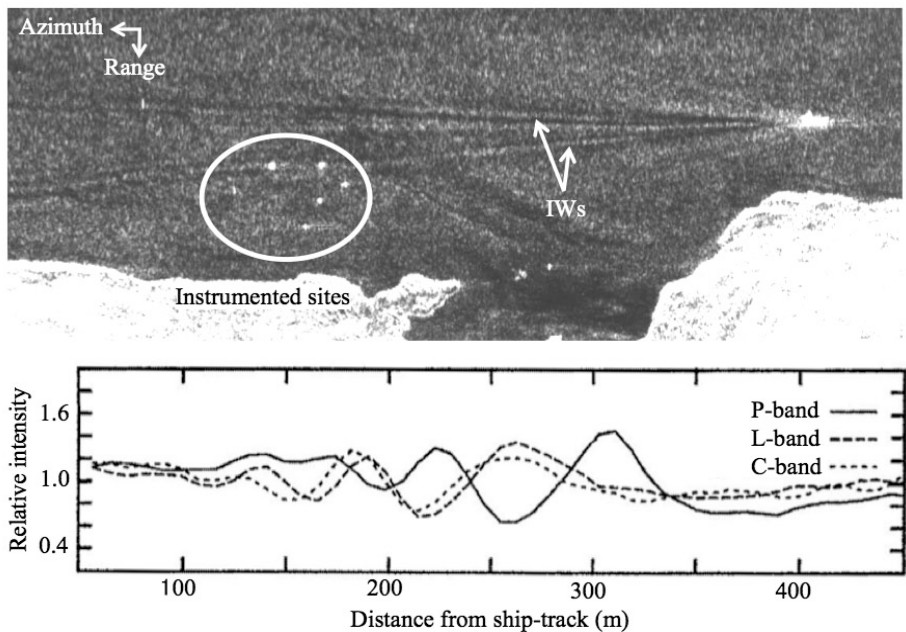

**Figure 13.** Upper: X-band SAR image of IWs and instrumented sites in Loch Linnhe, Scotland (5 × 2 km). Lower: Relative intensity modulation of multi-frequency AIRSAR images.

While most studies on oceanic IWs are on the basis of intensity, the first measurement of IWs by ATI SAR was made with the JPL (Jet Propulsion Laboratory) airborne ATI SAR during the 1989 Loch Linnhe campaign, where the line-of-sight Doppler velocity estimated by the JPL ATI SAR ranged from 50 cm/s to 100 cm/s. However, the in situ current data were 5 cm/s to 10 cm/s [88]. This large discrepancy is due to the different modulation by the Bragg waves advancing and receding from the SAR, in particular, under the wind direction across the IW features. This contribution needs to be subtracted from the Doppler velocity for accurate measurements of surface currents induced by IWs [89].

The first IW parameter retrieval by the spaceborne ATI SAR was reported using the TerraSAR-X dual receive mode (DRA mode) over the waters around the Dongsha (Pratas) Atoll in the South China Sea [90]. No in situ data were available, but using the previously reported information on the water depth, densities and thickness, the theoretical results based on the K-dV equation agreed fairly well with the observed Doppler velocity and image intensity signature.

Note that the TerraSAR-X DRA ATI mode consists of a single antenna for transmission and two antennas for reception of the backscattered signal separated in the along-track direction on the same antenna board. Due to the limited distance between the two receiving channels on a single antenna board, the system is suitable for high frequency bands such as X-band, and the TerraSAR-X DRA utilizes a 4.8 m long antenna with the receiving antenna separation of 1.2 m [91].

## 4. Oil Slicks

### 4.1. Background

Oil slicks are often caused by tanker and offshore accidents, and illegal discharge by vessels and plants through rivers, resulting in short- and long-term environmental and economic damage. For example, the 2010 Deepwater Horizon oil-rig explosion was the largest marine oil spill in history, discharging an estimated 4,900,000 barrels (7.33 barrels ≈ 1 ton)

of crude oil and 4 million pounds of gas into the Gulf of Mexico, killing 11 workers and doing considerable damage to the Gulf ecosystem [92–95].

Another cause is by tankers. Figure 14 shows the major oil spills from 1967 to 2021, where, except the Hebei Spirit and Sanchi cases, the largest spills occurred before the year 2000. From the 1990s to 2020s, there have been 602 cases involving tankers of 7 tons and over, resulting in 1,494,000 tons of oil spills. Most of the spills were due to collisions and groundings. It should be noted that the numbers of spills involving tankers are in a decreasing trend since the 1990s (358, 181, and 63 spills during the 1990s, 2000s and 2010s, respectively) [96].

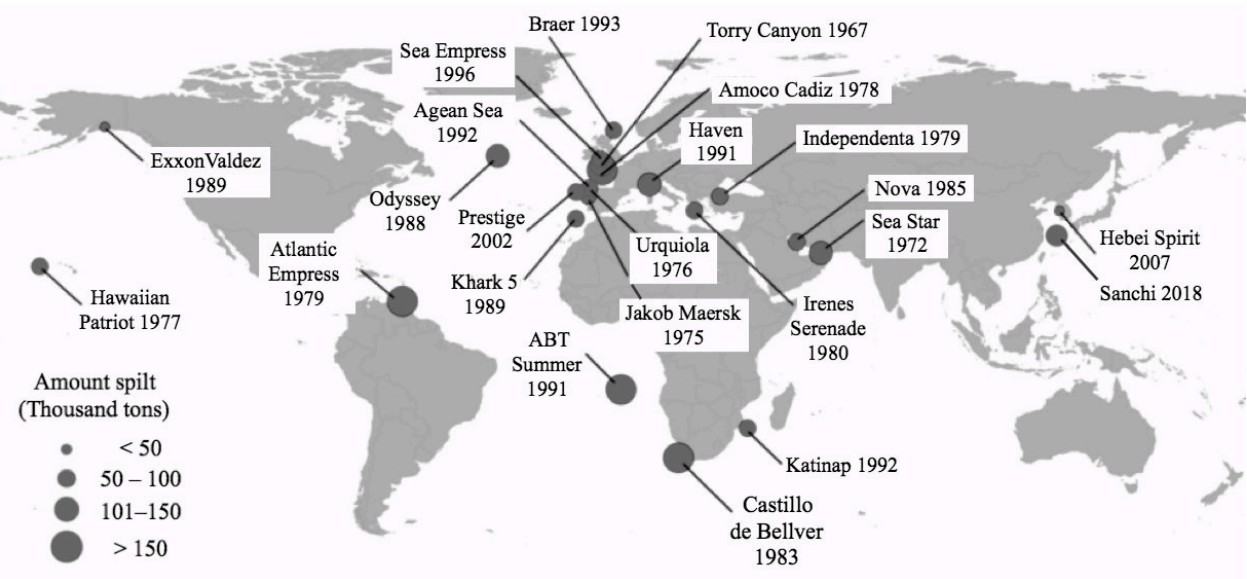

**Figure 14.** Major oil spills by tankers from 1967 to 2021. (Courtesy of ITOPF 2021 [96]).

One of the latest spills, among others, occurred when the anchored Hebei Spirit oil tanker collided with a crane vessel off the east coast of Korea on the 7 December 2007 [97]. Figure 15 shows the ENVISAR-ASAR C-band image acquired four days after the collision. In the figure, the dark areas around the tanker (white circle) correspond to the spilled oil. The other dark areas in the top-left and those near the coasts are considered as wind-sheltered areas. Similar dark images are observed, for example, by cold up-welling water, biogenic slicks, rain cells, and algae. These images are called "look-alikes" and the classification of oil spills and look-alikes is one of the main research subjects, as well as the classification of different types of oil such as crude oil, plant oil, emulsion (mixture of oil and water), and natural seep oil [98]. Note that the long white linear images are associated with the interaction between the currents and bottom topography, as will be discussed later.

The main task of SAR is the detection of oil spills, classification of different types of oil and look-alikes, evolution of slicks, and oil thickness.

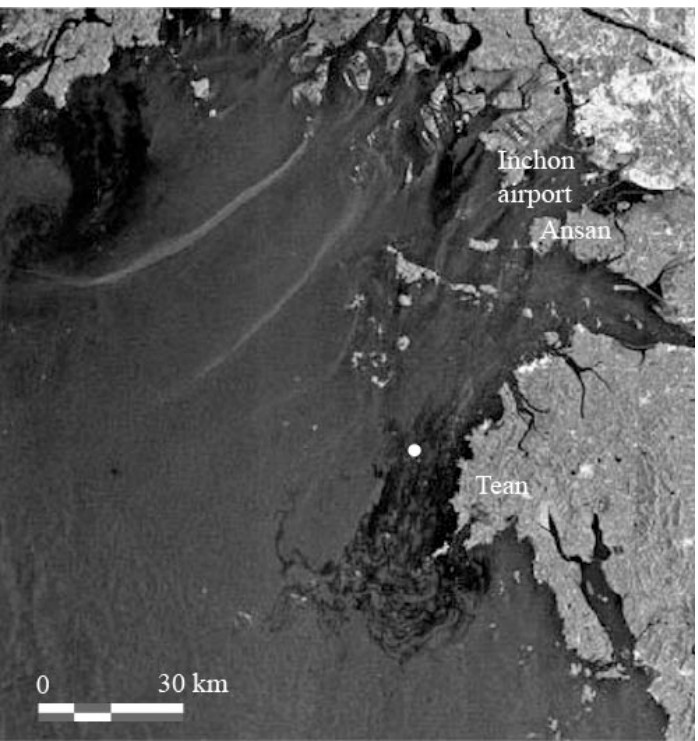

**Figure 15.** ENVISAT-ASAR C-band image, showing the Hebei Spirit tanker oil spill off the west coast of Korea. The dark areas around the position of the collision (white circle) is the spilled oil. The image was acquired 4 days after the collision.

*4.2. Imaging Process*

To date, the images of spilled oil have been acquired both by optical sensors and SAR [97], following the analyses based on the conventional intensity-based method, e.g., [99–102], combinations of SAR and optical data [100,103], polarimetry [37–39,104,105], and recent artificial intelligence [106–109].

The principal mechanism of oil slick detection using image intensity is through the change of surface roughness. The surface tension of oil is greater than water, so that a film of oil on the sea surface damps the small-scale Bragg waves. Then, the microwave incident on such a smooth surface is mostly reflected away in the specular direction, while there is sufficient backscatter from the surrounding rough clean surface. Thus, the images of slick areas appear darker than the clean surface, as in Figure 15. The degree of reduction in image intensity of mineral oil films is smaller (~3 dB) than that of biogenic films (~10 dB), while those of crude oil and emulsion are similar (~8–9 dB) [98].

In general, high frequency SARs such as X- and C-bands yield higher image contrast, since damping by oil slick increases with increasing Bragg wavenumber. Hence, the radar backscatter of high microwave frequency from the clean sea surface is larger than that of L/P-band microwaves, i.e., the difference in image intensity between oil-covered and clean surface is larger, e.g., [110]. For the same reason of strong radar backscatter, VV-polarization is suitable compared with HH-polarization in the incidence angles ranging from 30° to 60° [111]. Radar backscatter in HV/VH-polarization is due to multiple-bounce scattering, while the backscatter from the sea surface is single-bounce surface scattering. Hence, the radar backscatter from both the clean and oil surfaces is very small, i.e., very low image contrast. However, the controlled oil release experiment using two airborne SARs including the Uninhabited Aerial Vehicle SAR (UAVSAR) operated by NASA/JPL showed that the HV-polarization was recommended because of high slick-sea contrast under a sufficiently low system-noise floor [111].

Wind speed also affects the image contrast. Under little and no wind (≤2–3 m/s), both the oil and calm sea surfaces become equally smooth, so that the images appear

similarly dark (see the areas of little/no wind in Figures 1 and 15). Suitable wind speed is, in general, between 2–3 m/s and 10–14 m/s, and for natural light oil and heavy crude oil, it is ≤6–7 m/s and ≤10–14 m/s, respectively. Under very strong wind, the surfaces become equally rough, yielding a negligible effect of the surface tension of oil slicks. Evaporation, dispersion and dissolution of oil may also occur. Thus, the images of oil and clean sea surfaces become equally bright.

*4.3. Detection and Classification*

Figure 16 is an example of oil spills and look-alikes detected by different SARs, showing natural oil seeps, oil slicks from off-shore oil-rigs, low wind and up-welling look-alikes, and oil spills illegally discharged by a ship transmitting automatic identification system (AIS) signals at the same time. From these images, the detection and classification of the different types of oil spills can made, and Figure 17 illustrates the general procedure.

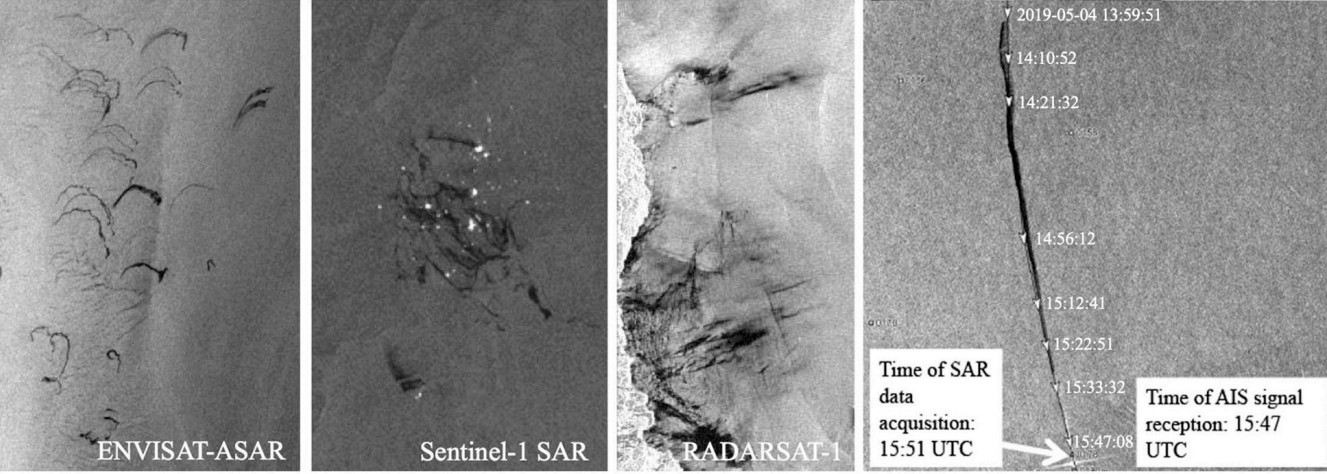

**Figure 16.** From left to right: Natural oil seeps, oil slicks from off-shore oil-rigs, low wind and up-welling look-alikes, and illegal oil discharged by a ship identified by the automatic identification system (AIS) signals.

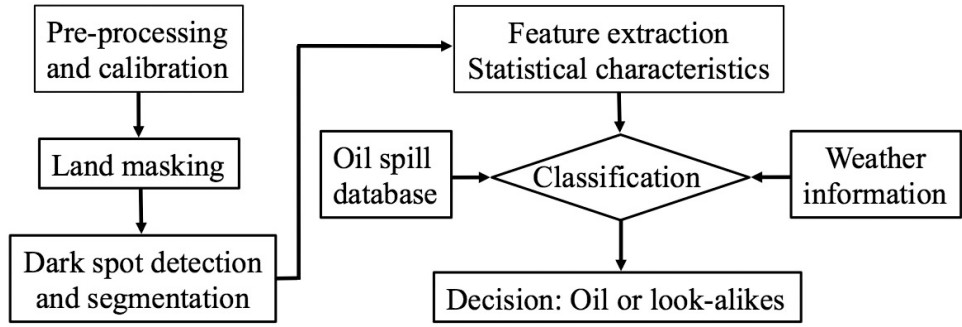

**Figure 17.** General procedure for oil slick detection system.

As in the figure, dark spot detection and segmentation are made after SAR data calibration and land masking. Then, the image features and statistical characteristics such as the probability density function (PDF) of image intensity are computed. Comparison of these characteristics is made with those from the oil spill database, and the decision is made whether the image is of oil slicks or look-alikes. In general, there are four main approaches based on the procedure shown in Figure 17 to the detection and classification of oil slicks as follows.

### 4.3.1. Manual Approach

The first is the manual approach using intensity images by trained operators who make decision with supporting information on ships with and without AIS, wind, current, location of oil-rigs, coastlines, national territorial borders, and chlorophyll, etc. Since the accuracy depends on the operators' skill and available information, this method is not exact science, and there exists an uncertainty associated with the results [112]. Nevertheless, this method may be practical by incorporating with other approaches.

### 4.3.2. Semi-Automatic/Automatic Approach

The second approach is the semi-automatic/automatic method using single-polarization intensity-based analyses. Classification of oil-covered areas from clean surface by image classifiers such as histogram threshold, Bayesian classifier, and target detection algorithms, such as CFAR (constant false alarm rate) and ATM (adaptive threshold method). The classification of oil and look-alikes is based on the features such as slick shape, brightness, surroundings and homogeneity. This method is often applied in practice since the basic intensity data are used with simple detection algorithms. For example, Solberg, et al. [102] used 59 RADARSAT and ENVISAT images to classify oil-spills over European waters. The method is the same as that in Figure 17. ATM with a moving window was used for dark spot detection. In the detection algorithm, the threshold value $I_T$ is defined as $I_T = <I>—c$, where $<I>$ is the mean intensity within the window and c is the empirical constant. Classification was made using the Bayesian classifier based on the probability density functions of oil, look-alikes, and the comparison was made by aircraft data and the manual detection by trained operators at the Oil Spill Detection Service of Kongsberg Satellite Service (KSAT), Norway [113]. The performance in detecting verified oil slick was 72% and 77% for the ENVISAT and RADARSAT data, respectively. Other operational systems include GNOME (General NOAA Operational Modeling Environment [114,115] and IMO (International Maritime Organization [116]), etc.

### 4.3.3. Polarimetric Analyses

The third approach is based on polarimetric analyses. Popular algorithms include span (HH/VV/HV mean power), four-component scattering model, $H/\alpha$ classifier, where H is entropy and $\alpha$ is alpha angle, and polarization power ratio [17–22]. The four-component scattering model decomposes the scattering power into surface, double-bounce, volume and helix scatterings using the covariance matrix of the HV-based scattering matrix. The entropy is derived by the eigenvalue analysis of the coherency matrix, and describes the randomness of the scattering process. It takes the values between 0 and 1. For example, $H = 0$ means one eigenvalue, indicating only one scattering process, such as the surface scattering from the sea surface, while $H = 1$ means three identical eigenvalues (in case of HH/HV/VV data), indicating the randomness of the scattering process, such as system noise. The alpha angle defines single-bounce ($\alpha = 0°$) and double-bounce ($\alpha = 90°$) scattering processes. $H/\alpha$ classifier is often used for land-cover classification.

As an example, the multi-polarization features were reported using the quad-polarization RADARSAT-2 C-band data with controlled oil spills (plant oil, crude oil, and emulsion) and simulated look-alikes (monomolecular biogenic film) in the North Sea [37]. The cross-polarization data were at the system-noise level, and not useful for the analyses. The mean VV-polarization signals were above the system-noise floor, but the mean HH-polarization levels lie close to and partly in the noise level. After comparisons of the in situ data with different polarimetric features such as entropy and inter-polarization correlation, the most useful parameters for discriminating oil and look-alikes were the geometric intensity (determinant of the coherency matrix) and the co-polarization ratio $<|S_{HH}|^2>/<|S_{VV}|^2>$, where $S_{HH}$ and $S_{VV}$ are the scattering matrix elements, i.e., image complex amplitudes of the corresponding co-polarizations. The co-polarization ratio is also a simple parameter to evaluate dielectric constants. It should be emphasized that radar backscatter from both the oil-covered and clean surfaces is the single-bounce surface scattering, while the polarimetry

is based on multiple-bounce scattering. Thus, these results, along with others [38,102,105], are considered as not associated with the scattering process but rather with the dielectric properties of oil, look-alikes and sea water, as indicated by the co-polarization ratio, and the system-noise level relative to the scattering power [98,117]. Another problem arising from the practical applications is that the polarimetric mode is not routinely operational.

### 4.3.4. Artificial Intelligence

The last and most recent approach is AI. This method can be applied to both the single-polarization and polarimetric data. Algorithms include the early versions of supervised classifiers such as SVM, standard/artificial NN, and wavelet neural network to CNN [106–109]. The method is accurate, but requires a large amount of training data.

At this stage, a brief summary is given on the deep learning method, since AI is also used for the waveheight measurement in Section 2.3, and IW detection in Section 3.2, as well as the detection and classification of ships. For target detection, the deep learning model usually detects target by bounding boxes, e.g., faster R-CNN, single-shot detection (SSD) [24,75,118] and you only look once (YOLO) family [119–121]. For classification, the model classifies pixels into different categories, called semantic segmentation (e.g., U-Net) [122]. The combination with detection and classification is called instance segmentation (e.g., Mask R-CNN) [123]. As an application for SAR images, detection and/or classification can be used for ships, oil spills, internal waves, sea ice, etc. The detection can provide the number, size, and location of targets. Different types of targets can also be detected by training with each supervised category. Meanwhile, the semantic segmentation model can yield the area of targets, and it can grasp the occupied areas or concentrations of the targets in SAR images.

The early version of deep learning consists of a single hidden layer and training data such as the features of geometry and shape, features of mean intensity, standard deviation of dark spots, and the ratios of those of clean sea surface, and texture, etc. Singha, et al. [124], for example, used 183 oil spots and 720 look-alikes in 92 ERS-2 and ENVISAT-ASAR images for training NN over European waters, and 226 oil spots and 4670 look-alikes in 82 ASAR and RADARSAT-2 images for the test data. All the data were confirmed and recorded by the European Space Agency (ESA) and European Maritime Safety Agency (EMSA). The results showed high classification accuracy of ~92% and ~98% for oil slicks and look-alikes, respectively. Zhang, et al. [125] used the same RADARSAT-2 polarmetric data with a controlled oil spill experiment with the ground-truth data [37], and compared the classification accuracy of NN, support vector machine (SVM) and maximum likelihood classifier (MLC) methods. The features used include intensity, and polarimetric parameters such as entropy, alpha angle, correlation coefficient, etc. The detection accuracy of oil spills was above 95% by all classifiers.

CNN is mainly used for image analysis, consisting of multiple hidden NN layers of convolutional, pooling and fully connected layers between the input and output layers, as shown in Figure 18 [106,108]. The convolution layer can catch the characteristics of the target. The process of convolution involves a filter checking if a feature is present. The pooling layer also rolls a filter across the input image. It reduces the number of parameters in the input. It means the increasing robustness of the model. By accumulating such layers, the deep neural network is realized. The fully connected layer is the roll of image classification based on the features extracted in the previous layers. Like this, CNN can identify targets or features of an image. To obtain high performance, training requires a large quantity of data, but too much training may result in over-fitting. Generally, the model is set to avoid over-fitting and remains suitable for practical applications. Further recent examples are illustrated as follows.

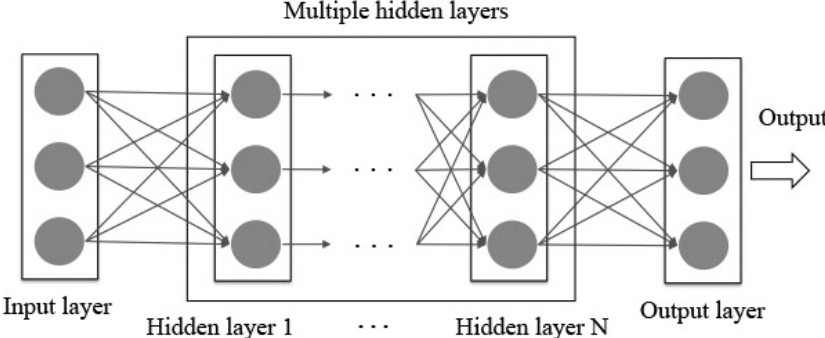

**Figure 18.** Deep learning consisting of input, multiple hidden neural network layers, and output layer.

A feature merge network (FMNet) model was developed to distinguish between oil-spill areas and oil-spill-like areas [126]. 1002 images were used for training and 110 images for testing, with five classes (oil spills, look-alikes, land, ships, and sea areas). The initial SAR data were collected from the ESA database, the Copernicus Open Access Hub, acquired through the Sentinel-1 European Satellite missions. The required geographic coordinates and time of the confirmed oil spills were provided by the European Maritime Safety Agency (EMSA) based on the CleanSeaNet service records, from 28 September 2015 to 31 October 2017. The overall accuracy reached 61.90% compared with U-Net. The recognition accuracy of oil spill areas and oil-spill-like areas reached 56.33%.

A novel automatic detection of offshore oil slicks was developed using multi-modal deep learning [127]. An expert photo-interpreter annotated Sentinel-1 SAR data over four years and three areas, specifically, the Atlantic Ocean coast in Southern Africa (Nigeria and Namibia), and the Mediterranean Sea in Western Asia (Lebanon). It utilized a refined version of the FC-DenseNet model, a well-known extension of densely connected convolutional networks (DenseNets). The proposed approach reached a detection performance of up to 94%.

As an example of an application of polarimetric SAR, an improved deep learning model named BO-DRNet was proposed [128]. In this model, target features could be extracted more efficiently than the conventional model using ResNet-18 as the backbone in encoder of DeepLabv3+ [129], and Bayesian optimization was used to optimize the hyper parameters. In the study, the analyses were made with three sets of quad-polarimetric oil spill SAR images acquired by RADARSAT-2 over the Gulf of Mexico, and 10 prominent polarization features were extracted from these images. Then, 32,768 pixels of oil spills and 32,768 non-oil spill pixels were trained and tested. The other models such as FCN-8s, DeepLabv3 + Xception and DeepLabv3 + ResNet-18 were used for comparison. As a result, BO-DRNet performed best with a mean accuracy of 74.69% compared to the other models.

As a last example, Faster R-CNN was applied for the detection of oil spills with C-band Sentinel-1A/B (S1-A/B) and RADARSAT-2 (RS-2), over the Gulf of Mexico, the Indian Ocean, and the East and South China Sea [130]. VV-polarization was used since it has a better signal-to-noise ratio than other polarizations. A large data set, consisting of 15,774 labeled oil spills from 1786 C-band Sentinel-1 and RADARSAT-2, was used to train, validate and test. The results showed a mean precision of 92% for the detection of oil spills with wide swath SAR images.

### 4.4. Tracking and Prediction

After oil is released, evaporation, dispersion, and dissolution may occur, as well as diffusion. The tracking and prediction of oil spill movement are important for prompt measures of prevention and cleaning. There are several oil diffusion models, such as those of GNOME by NOAA [114,115], IMO model [116], and ROMS [97,131]. The principle of forecasting the oil slick movement is illustrated in Figure 19. After detecting oil slick areas at time $t_1$, the SAR image is gridded. Cells containing oil slicks are called virtual oil

particles. Then, the empirical diffusion equation $\mathbf{V}_{oil} = \mathbf{V}_{current} + Q \bullet \mathbf{V}_{wind}$ is applied to predict the positions of oil particles at a later time ($t_2$). In the equation, $\mathbf{V}_{oil}$, $\mathbf{V}_{current}$, and $\mathbf{V}_{wind}$ are velocity vectors of oil, current and wind, respectively, and the wind-induced current velocity is expressed in terms of the wind drift factor $Q$. The wind drift factor can range from 0.01 to 0.06 [131–135], but in many cases, $Q$ is fixed as 0.03 [116,136]. Information such as the evaporation and emulation can be included if the data are available.

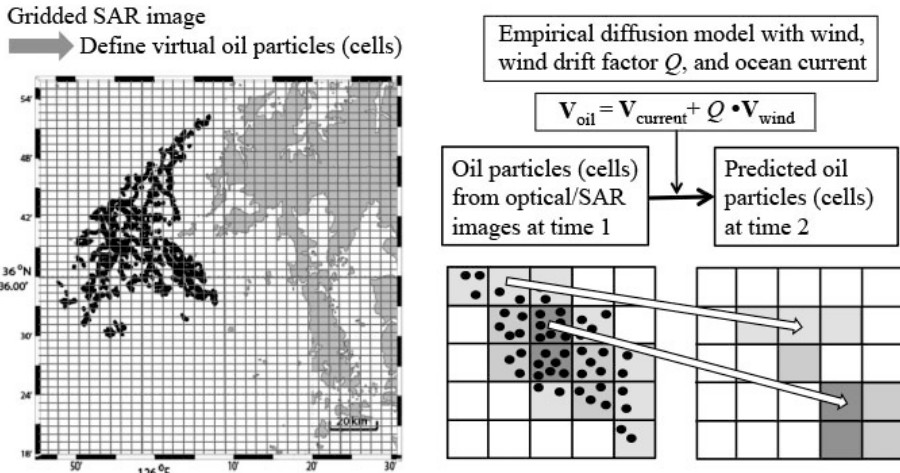

**Figure 19.** Prediction of oil movement, where $\mathbf{V}_{oil}$, $\mathbf{V}_{current}$, and $\mathbf{V}_{wind}$ are the velocity vectors of oil, current and wind, respectively. $Q$ is the wind drift factor.

Figure 20 shows the case of the oil prediction for the *Hebei Spirit* tanker accident [137]. The accident occurred at local time 07:15 on 7 December 2007 off the east coast of Korea [97]. After the accident, airborne sensors as well as spaceborne sensors acquired the data, including the KOMPSAT-2 optical data (11:04/08), ENVISAT-ASAR (10:40/11), RADASAT-1 (18:31/11), TerraSAR-X (06:44/13), and ENVISAT-ASAR (10:45/14). First, the oil particles were detected using the KOMPSAT-2 image, and the positions of oil particles were simulated at the time of the next data acquisition by ENVISAT-ASAR. For the simulation, the Environmental Fluid Dynamics Code (EFDC) and automatic weather system (AWS) were used to generate the tidal and wind fields, respectively. The simulation was repeated for different $Q$ values in order to find the best match with the ENVISAT-ASAR oil particles. Figure 20 illustrates an example for the simulation of the RADARSAT-1 oil particles from the ENVISAT-ASAR, and the best fit was found with $Q = 0.015$, with a matching accuracy of 60% to 86%. On average, the best $Q$ value was 0.031. Thus, it seems that the fixed $Q$ value of 0.03 may be a good choice, but it depends on the wind speed as in the top-right of Figure 20, as well as the current.

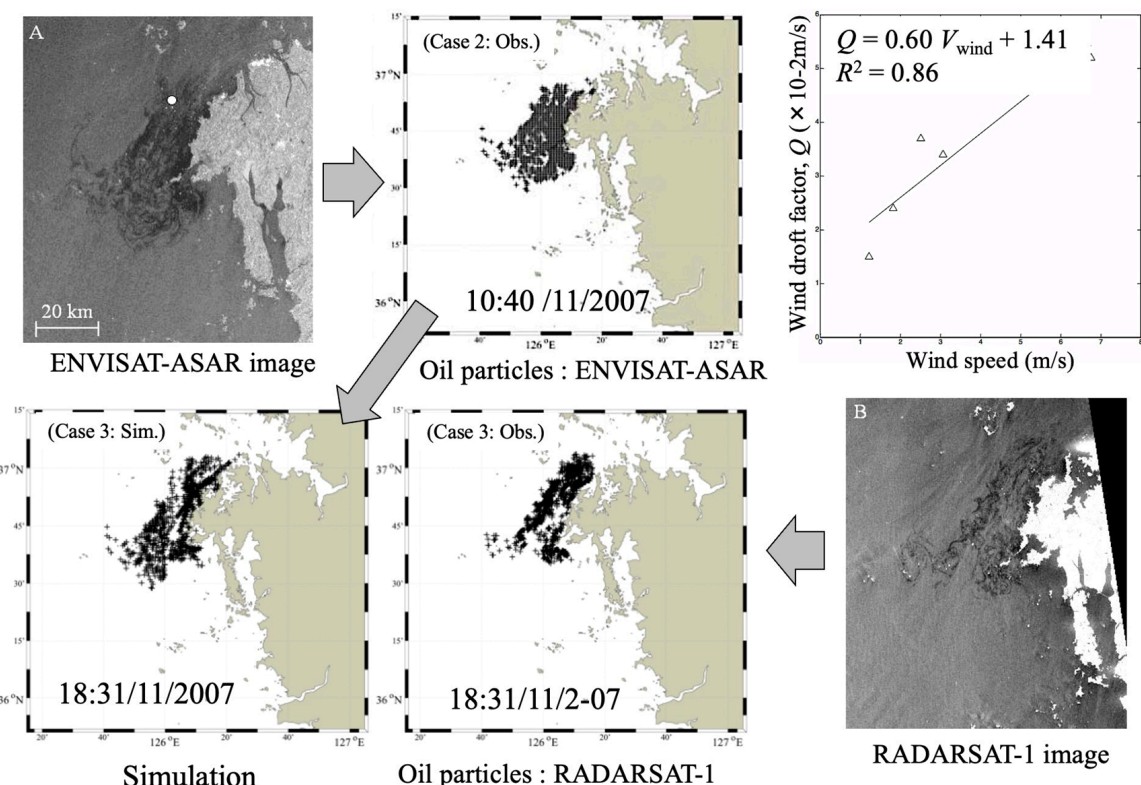

**Figure 20.** (**A,B**) An example of the prediction of oil slick movement for the *Hebei Spirit* tanker oil spill off the west coast of Korea shown in Figure 15. Oil particles are extracted from the ENVISAT-ASAR image and simulated and compared with the oil particles in the RADARSAT-1 SAR image. The top-right image shows the optimum wind drift factor in terms of wind speed [137].

The oil diffusion model can also be used to track back the unknown sources of oil spills [138]. The sources can be identified if they are visible in SAR images, such as oil-rigs, wrecked ships, natural oil seeps of characteristic shapes, and ships with AIS signals, as in Figure 16. Note that the ships discharging waste oil illegally do not, in general, transmit the AIS signals simultaneously. Matching of AIS signals with the AIS transmitting ships is a subject of ship detection and identification [139].

## 5. Ocean Current

Ocean currents have a strong impact on the global climate through heat and nutrients circulation as well as fishing activity. In SAR images, the current boundaries can be visualized by the changes of intensity, such as the front in Figure 1. Apart from the qualitative estimation of current boundaries, the measurements of current velocity are made using the Doppler shift of return signals and ATI SAR. The complex SAR data can be used for the velocity measurement of ocean currents through the Doppler centroid shift of the azimuth spectra, as shown in Figure 21.

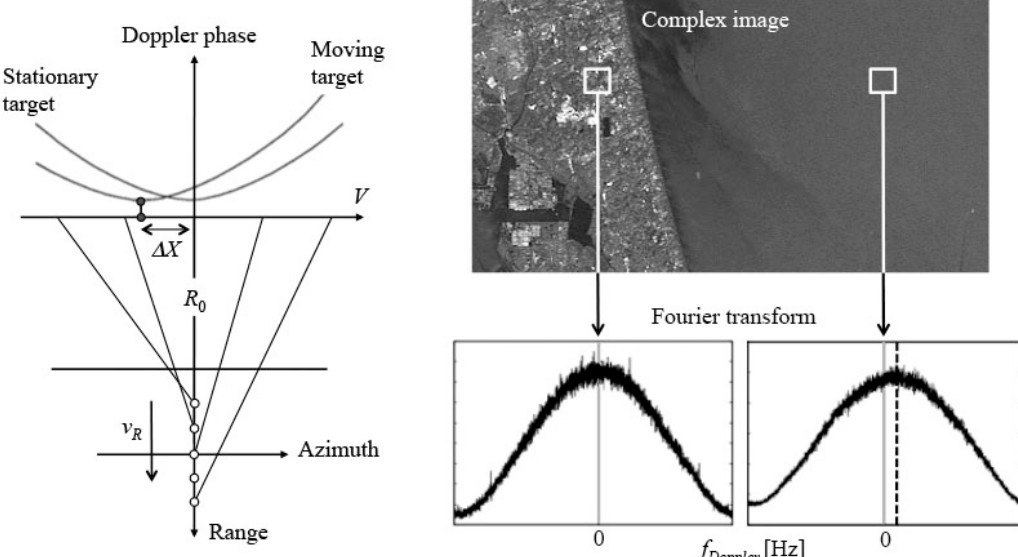

**Figure 21.** Principle of current velocity measurements by the Doppler shift. **Left**: the center of the azimuth return signal from a range moving scatterer is shifted. **Right**: The center of the spectrum over the sea surface is shifted from that of the stationary surface.

As noted in Section 2.3 on breaking waves, the image of a scatterer moving with the ground-range velocity $v_R$ is shifted in the azimuth direction by a distance $\Delta X = -(R_0/V)(v_R \sin\theta_i)$, where $R_0$ and $V$ are the slant-range distance and platform velocity, respectively, as in the left of Figure 21. Due to the same reason, the center of the Doppler spectrum, computed by Fourier transform of the image complex amplitude of the sea surface, is shifted. The shift is proportional to the range velocity of ocean currents, intrinsic velocity of the Bragg waves, and the orbital velocity of waves if present; ocean wind also takes an important role. This method has been used for the current measurements by several researchers since 2005 [140–143].

The ocean current measurement by ATI SAR was first proposed in 1987 by Goldstein and Zabker [144]. As mentioned in Sections 1 and 3.3, ATI SAR consists of two antennas in the along-track direction on airborne or spaceborne SARs. The geometry is shown in the top-left of Figure 22, where the two antennas are separated by the distance $B_A$, and the aft antenna is assumed for signal transmission and reception while the fore antenna is for reception only. Since the time difference when the fore and aft antennas are directly above the scatterer is $B_A/(2V)$, the difference of the slant-range distance is $(v_R \sin\theta_i)B_A/(2V)$. The complex interferogram is given by $A_F A_A{}^* = |A_F| |A_A| \exp(i\delta\phi)$, where $A_F$ and $A_A$ are the complex amplitude through the fore and aft antennas, respectively. The interferometric phase is $\delta\phi = k(B_A/V)v_R \sin\theta_i$, where $k = 2\pi/\lambda$ is the wavenumber and $\lambda$ is the wavelength. The phase value is relative, and to compute the absolute phase, a stationary reference point is required. Note that there is a phase ambiguity if $\delta\phi$ exceeds $2\pi$. For a C-band ($\lambda = 0.057$ m)) airborne ATI SAR with $V = 200$ m/s and $B_A = 0.6$ m, the measurable velocity without the phase ambiguity is 19 m/s. Figure 22 shows an example of the ocean current measurement by ATI SAR across the Gulf Stream. The accuracy was found as ~0.1 m/s.

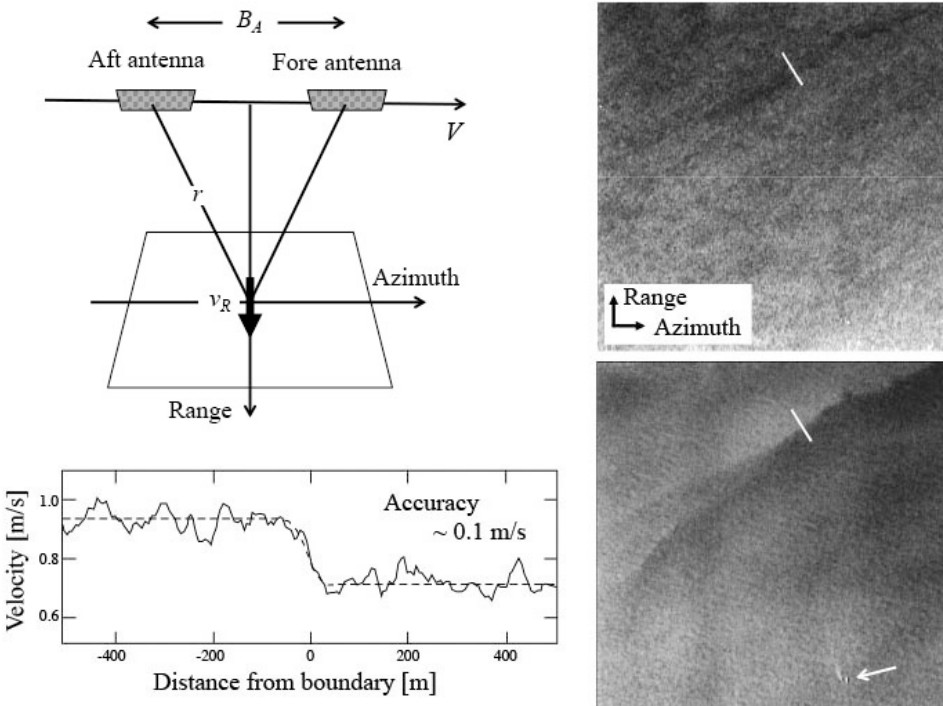

**Figure 22.** From top left in clockwise direction: Geometry of ATI SAR, intensity image (8 km square) of the Gulf Stream boundary off Virginia Beach, USA, produced by the JPL-AIRSAR, InSAR phase image, and estimated current velocity. The survey vessel (arrowed) is used as a reference to compute the absolute interferometric phase and the current velocity [145,146].

There have been several airborne ATI SAR, e.g., [147–149], and spaceborne systems including ATI SAR by the space shuttle [150], the dual-channel RADARSAT-2 MODEX-1 (Moving Object Detection Experiment) mode (two signal receiving channels separated in the along-track direction on a same antenna board) [151], and the dual-channel and formation flight of TerraSAR-X and TanDEM-X [11–13,152]. These systems can measure only the cross-track (range) component of the current velocity.

For current vector estimation, the dual-beam ATI SAR was reported by using a pair of antennas, with one squinted forward and the other squinted backward [153]. The University of Massachusetts developed an airborne ATI SAR consisting of four antennas with the two forward-looking antennas and the other two backward-looking [154]. Wollstadt, et al. [14] presented a Ku-band spaceborne ATI SAR scheme consisting of three antennas. The main antenna at the center transmits forward- and backward-beams, and the two antennas separated by 12 m receive the return signals. Ouchi, et al. [15] proposed a multi-aperture ATI SAR (MA-ATI SAR) using the conventional ATI SAR data for the current vector measurements with supporting simulation and airborne data [16,155]. In this MA-ATI SAR, two-look processing is applied to the data acquired by the fore and aft antennas, yielding the forward- and backward-looking complex image amplitudes, and the corresponding velocity components are produced from the interferograms in each look direction. The spaceborne ATI SAR for the current vector measurements is a subject of future study.

## 6. Bathymetry

The white linear images in the upper-left in Figure 15 are associated with the current interaction with the topography of the sea floor [156]. The bottom topography is important information for coastal navigation, offshore construction, aquaculture, and climate change associated with the rising sea levels and current circulation [157]. While the bathymetry measurements, in general, are carried out by survey vessels, the approach using SAR data was first proposed in 1983 [158]; the principle is illustrated in Figure 23.

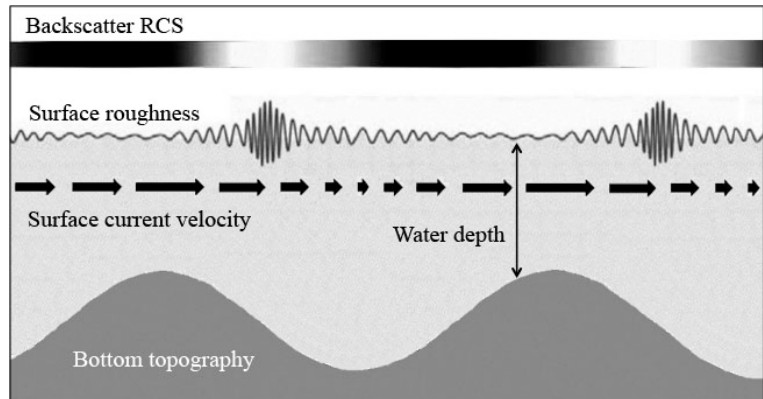

**Figure 23.** Illustrating the dependence of the backscatter radar cross section (RCS) through the surface roughness changes caused by varying surface currents.

As the current passes above the undulating bottom topography, the velocity of the surface current varies depending on the depth. The relation between the current velocity and the water depth can be described by the sand wave model given by $V_{\text{current}} = C_{\text{onstant}}/d.$, where $d$ is the water depth. Due to the varying current velocity, the surface becomes rough in the current converging areas, yielding large backscatter RCS compared with the smooth surface of current converging areas. The surface roughness dependence on the converging and diverging currents is the same as for the case of internal waves in Section 3.

There are three main approaches for the measurement of bottom topography with SAR data. The first is based on the relation of the backscatter RCS and bottom topography. In the second method, the relation of the water depth and current velocity directly estimated by ATI SAR, and the third method uses the ocean wave images of wavelength and period related to the water depth through the dispersion relation.

An example of the RCS-based method is illustrated in the left of Figure 24. The surface current is first simulated from the initial water depth estimated by echo sounding. Then, the action balance equation with the current and wind data used in Section 3.3 for internal waves is applied to compute the waveheight spectrum. Given this simulated waveheight spectrum, the RCS is computed using the Bragg scattering model for comparison with the SAR image. This process is repeated for minimizing the difference between the simulated and SAR images, yielding the optimum water depth. This is the basis of the bathymetry in shallow waters by SAR that follows. The validation experiment of the operational system bathymetry assessment system (BAS) by ERS-1 SAR over the Waddenzee in northern Netherlands showed the root mean square error (RMSE) of 10–30 cm in comparison with the echo sounding data in the depth 2–8 m [159]. The initial version of BAS was one-dimensional, and a two-dimensional model was developed by BMT ARGOSS of the Netherlands [160]. See also a comprehensive summary of BAS in [161].

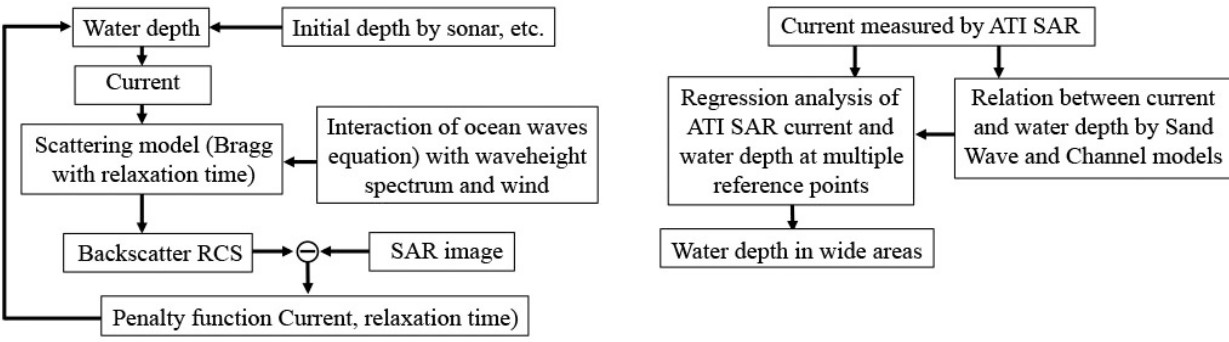

**Figure 24.** Flowchart for the bathymetry measurements by the RCS-based method (**left**) and ATI SAR-based approach (**right**).

In the ATI SAR-based approach, the current is directly measured by ATI SAR [149,152]. Since this method does not rely on the scattering model, the accuracy is considered to be higher than the RCS-based model. As in the right of Figure 24, the current velocity is estimated first by ATI SAR, followed by the regression analysis of the relation of surface current and water depth using multiple reference points. An example is illustrated in the left of Figure 25, where (a) the ATI-derived current vector to the north of the German island of Sylt, (b) water depth estimated by echo soundings, (c) 78 reference depths around the edge of the image, and (d) resultant depth map by the ATI SAR-based method [152].

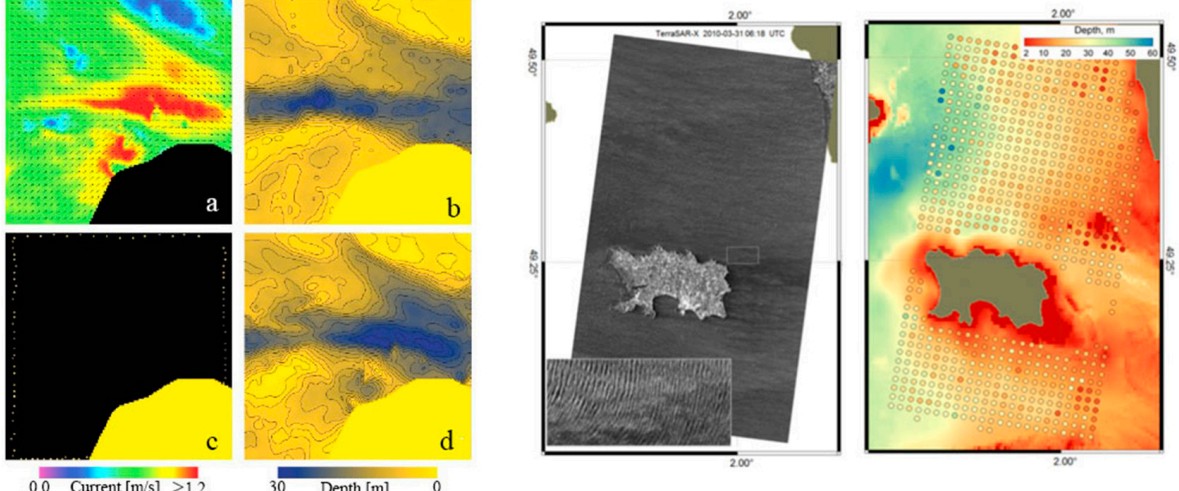

**Figure 25. Left**: (**a**) The current velocity vector measured by the X-band ATI SAR to the north of the German island of Sylt (3.5 km square). (**b**) Depth map from echo soundings. (**c**) 78 selected reference depth points (white dots). (**d**) Depth map derived in combination of the reference depth and the ATI-derived current field [152]. **Right**: TerraSAR-X StripMap scene over the island of Jersey in the southern exit of the English Channel (**left**), and the estimated bathymetry from the dispersion relation of ocean waves is shown at the points over the bathymetry data by the European Marine Observation Data Network [162,163].

The third approach is based on the dispersion relation of ocean waves and water depth given by

$$\Omega^2 = (gK + \gamma K^3)\tanh(Kd) \tag{1}$$

where $\Omega = 2\pi T_0$ is the angular frequency of ocean waves, $T_0$ is the wave period, $g = (9.81\ \text{m/s}^2)$ is the gravitational acceleration, $\gamma$ (~$0.735 \times 10^{-4}\ \text{m}^3/\text{s}^2$) is the surface tension/water density, $K = 2\pi/L$ is the wavenumber, and L is the wavelength. The wavelength can be measured from the SAR images, and the water depth $d$ can be estimated if the wave period $T_0$ is known. The wave period can be approximated as $T_0 = (2\pi L/g)^{1/2}$ under the condition $L >> d$, or depth data at reference points. There have been several reports on bathymetry measurements using ocean wave patterns [162,164–166], and an example is shown in the right two figures in Figure 25. The image was acquired by the TerraSAR-X StripMap mode over the island of Jersey in the southern exit of the English Channel (left), and the estimated bathymetry from the dispersion relation of ocean waves is shown at the points over the bathymetry data by the European Marine Observation Data Network (right) [162,163]. The RMSE is 7.1 m for this case, and the error increases towards deeper waters with longer wavelengths. Similar results based on the wave-based approach were reported using the Sentinel-1A C-band SAR off the Portuguese west coast, showing the relative error of the water depth ranging between 6% and 10%, but increasing with increasing water depth [166].

The effect of bottom topography to the surface currents is limited to the waters of depth less than 3–40 m. The depth of shallow waters in the English Channel shown in Figure 1 is 3 m to 33 m according to the sea chart [167].

## 7. Ship Detection and Classification

### 7.1. Background

The first study on ship detection by spaceborne SAR was reported after the launch of the SEASAT satellite in 1978 [168]. Since then, considerable effort has been made for its importance in maritime affairs, including ship navigation support, monitoring of fishing activity, detection of distressed ships, illegally operating ships such as pirates, smuggling and oil dumping, and also others associated with maritime domain awareness (MDA) [169].

There are several systems for the current ship navigation support, including AIS, Vessel Monitoring System (VMS), Long-Range Identification and Tracking (LRIT) and Vessel Traffic Service (VTS). AIS is mandatory for all passenger ships and ships over 300 tonnages to exchange information among cruising ships and ground-based monitoring stations on the identification codes, positions, courses, etc., directly or through satellite communication. Kongsburg (Norway), for example, operates 3 AISSat, the nano-satellites, sized 20 × 20 × 20 cm, mainly over European waters [170], and the exactEarth (Cambridge, ON, Canada) operating 18 AIS satellites covering the entire globe [171]. VMS is required for all fishing boats longer than 15 m in the European Union for monitoring the ships' information as well as the amount of catch through such communication satellites as Immarsat, ORBICOM and Iridium. LRIT monitors passenger ships, cargos, and mining ships through satellite VHF-band communication, and VTS is the system for ship traffic in and out of bays and ports through control radars, CCTV and wireless phones. As noted above, the problem is the ships that are not equipped with these systems and those not transmitting AIS and VMS signals. Thus, airborne and spaceborne SARs of all-weather and day-and-night data acquisition capability are important for ship monitoring.

### 7.2. Ship Detection

Figure 26 illustrates the microwave incident and backscatter from a ship on the water. The backscatter from the sea surface is due to the single-bounce surface scattering (apart from white caps from breaking waves), while there are several different scattering processes from the ship's structure, including surface, double-bounce, volume and helix scatterings. Among these scattering processes, double-bounce and volume scattering are dominant for those from ships, and thus the return signal power at cross-polarization is large compared with that of surface scattering from the sea surface. The scattering power from the sea surface is also smallest at cross-polarization in comparison with co-polarization (HV < HH < VV). The suitable polarization for ship detection is therefore cross-polarization. Figure 27 shows the ALOS-PALSAR PLR image of Tokyo Bay, Japan. The contrast of ships' images in white dots is clearly much higher in HV-polarization than co-polarization images.

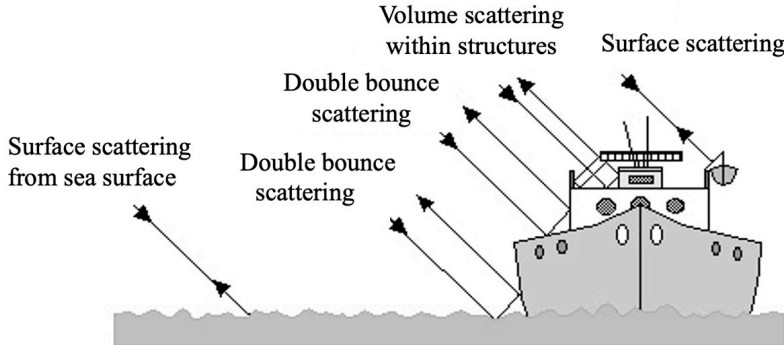

**Figure 26.** Illustration of the microwave backscatter from the sea surface and a ship. While the backscatter from the sea surface is dominated by the single-bounce surface scattering, the backscattering from the ship is due to the double-bounce, volume and helix scattering as well as surface scattering.

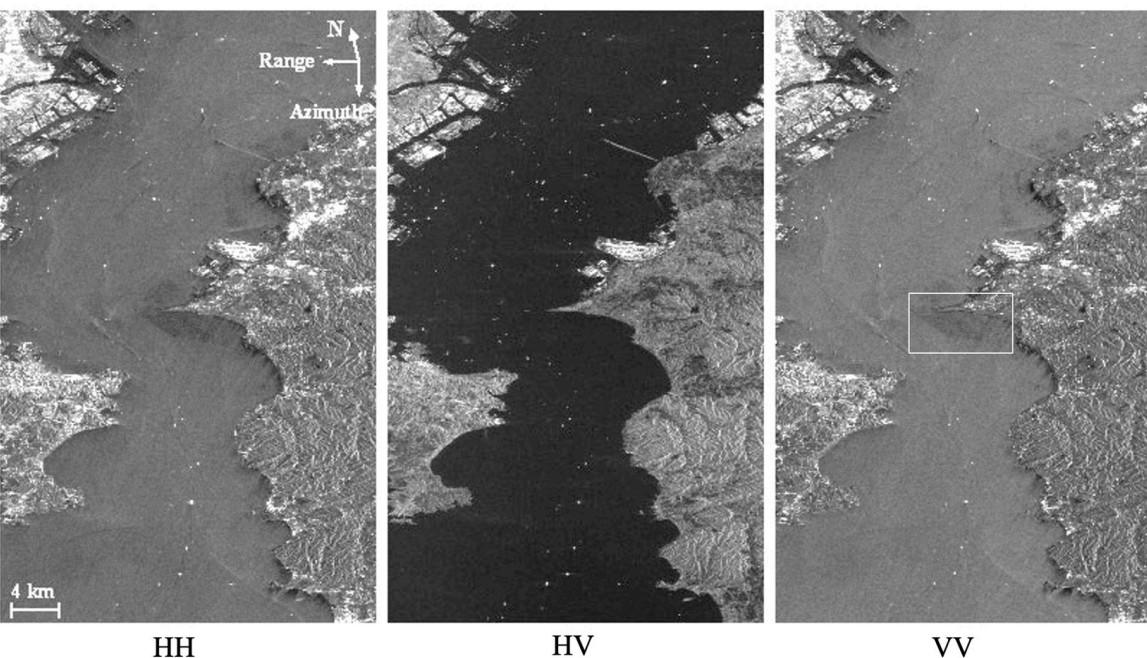

**Figure 27.** ALOS-PALSAR PLR images of Tokyo Bay, Japan, showing HH-, HV- and VV-polarizations from left to right. The image size is approximately 44 km and 28 km in the azimuth and range directions, respectively. The white rectangular box indicates the test site for detection of aquaculture.

For ship detection, many algorithms have been developed to date, e.g., [172–174]. Some known algorithms include constant false alarm rate (CFAR) of different versions [175–181], adaptive threshold method (ATM) [182,183], wavelet transform [184,185], multi/sub-look-based methods [186–191], polarimetric analysis [30–32], standard deviation filter (SDF) [192] and AI-based methods [41–45,118–121,193].

### 7.2.1. Constant False Alarm Rate (CFAR)

The principle of CFAR is illustrated in Figure 28. For detecting bright spots against the background noise, thresholding of image amplitude/intensity is applied using a moving window. In general, the background image amplitude often varies from an area to areas. If the threshold value is set as constant, the false alarm rate varies depending on the areas. In order to achieve a constant false alarm, a moving window is divided into a signal test cell, buffer window and background reference window. The size of the test cell is a single cell or larger, and decides whether the target is within the cell. The buffer window is not used, and should be larger than the expected size of ships. This prevents a part of the target being in the reference cell. The background reference window is used to compute the statistical distribution within the window.

In the cell-averaging linear-CFAR, for example, the clutter (noise) model that fits best to the data is assumed or sought by, for example, the Akaike information criteria (AIC) [194,195]. Then, the mean value within the reference window is calculated, and the pixel value in the test cell is divided by the mean, yielding the output that is independent of the parameters of the clutter. By setting a threshold for the required false alarm rate, the target can be detected with a constant false alarm rate.

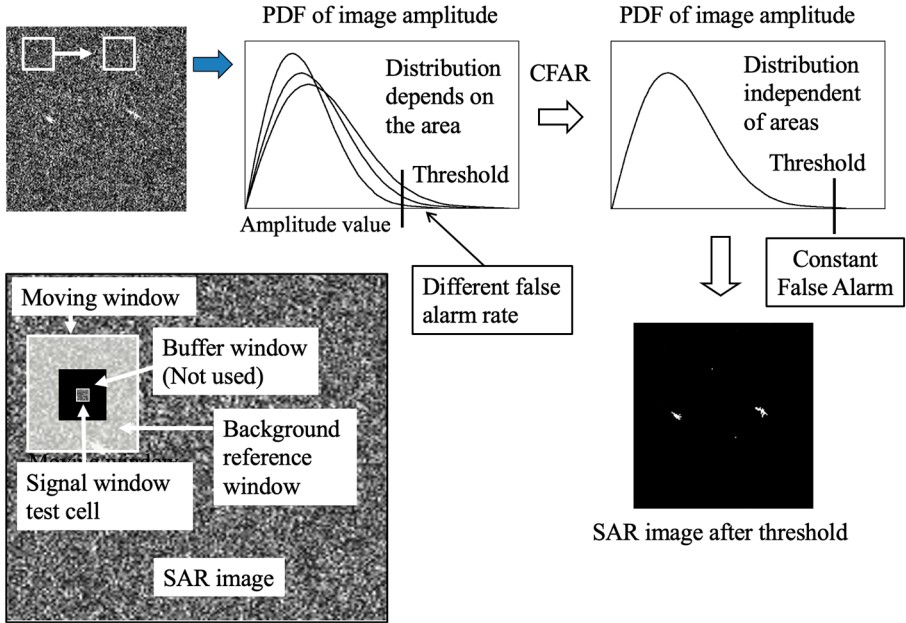

**Figure 28.** Illustrating the principle of CFAR for ship detection.

Consider the clutter with Rayleigh distribution with the probability density function (PDF)

$$p(z) = (2\,z/\sigma_z{}^2)\exp(-z^2/\sigma_z{}^2) \tag{2}$$

where $z$ and $\sigma_z$ are the image amplitude and standard deviation, respectively. The mean $<z>$ of the pixels in the reference cell is given by integrating $z\,p(z)$ over zero to infinity, which is $<z> = \pi^{1/2}\sigma_z/2$. Using the parameter $z' = z/<z> = 2\,z/(\pi^{1/2}\,\sigma_z)$, and from the relation, $p(z)dz = p(z')dz'$, the PDF is given by

$$p(z') = (\pi z'/2)\exp(-\pi z'^2/4). \tag{3}$$

which is independent from the clutter parameter $\sigma_z$. For the threshold value $z_T$, CFAR $p_{far}$ is given by integrating $p(z')dz'$ over $z_T$ to infinity, that is, $\exp(-\pi z_T{}^2/4)$.

Note that with increasing SAR resolution, the images of the sea surface contain statistically non-uniform textures such as those of white caps in high sea states. Then, the distribution of clutter tends to follow a non-Rayleigh distribution, such as the log-normal, Weibull, and K-distributions. Currently, the algorithms based on non-Rayleigh CFAR are used in most ship detection applications. Different versions of CFAR have also been developed, including GO/SO (greatest/smallest of)-CFAR, exclusion-CFAR, OS (order statistic)-CFAR, super pixel-CFAR, and others [175–181].

### 7.2.2. Adaptive Threshold Method (ATM) [182,183]

ATM uses the same moving window as for CFAR, but does not require the PDF of the clutter noise. The threshold $z$ is computed according to the equation, $z_T = <z> + c \cdot \sigma_z$ or $z_T = c\,(<z> + \sigma_z)$, where z can be amplitude or intensity in the reference window, and c is a constant. The empirical constant c depends on the data, and $c = 2$–$3$ are generally used. ATM is simple and easy to apply but the false alarm rate is not constant.

### 7.2.3. Wavelet Transform [184,185]

Among several different discrete WTs, the simple Harr wavelet is used for ship detection. As illustrated in the left of Figure 29, the principle is the addition and subtraction of neighboring pixels. The image after Harr transform consists of the mean value over $2 \times 2$ pixels in the original image, the mean values of difference between neighboring pixels in the horizontal, vertical, and diagonal directions. These four sets of values are added to

form a single output image of half the original image size. Thus, the process is similar to the edge detection in three directions. While the ATM is a single pixel-based method, WT is based on the average of 2 × 2 pixels, so that the target ship size is larger than that of the ATM. The right image in Figure 29 shows the COSMO-SkyMed X-band HH-polarization SpotLight image of Tokyo Bay, Japan. Most of the ships were detected and identified by AIS and visual observation by the National Defense Academy (NDA). Small spots detected in the upper end were not confirmed either by AIS or visual observation.

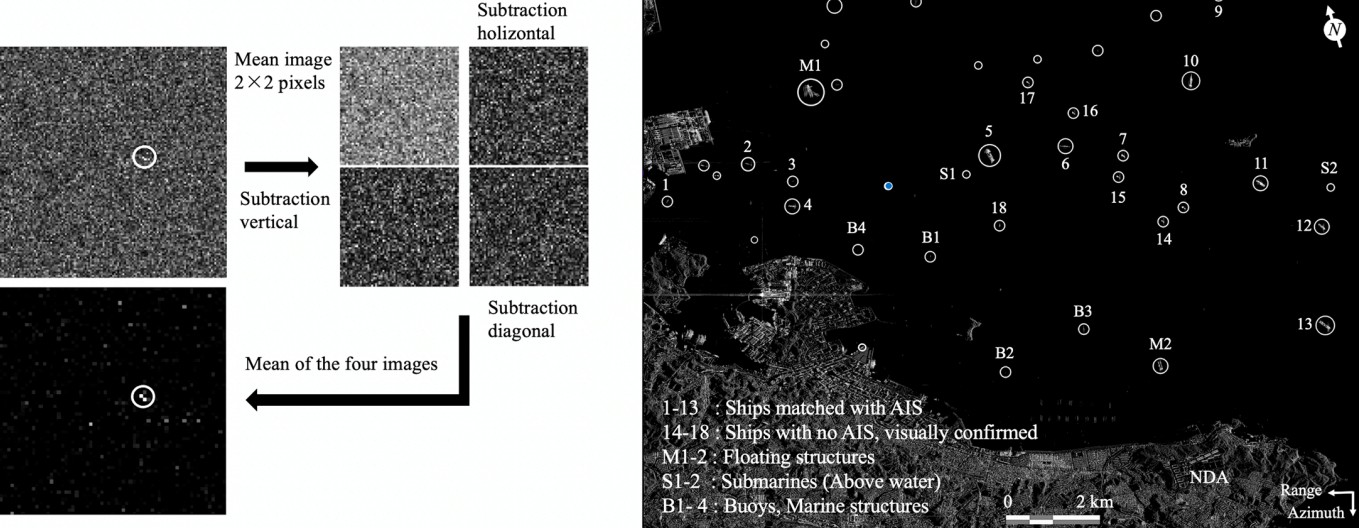

**Figure 29. Left**: Illustration of the principle of ship detection by the wavelet transform. **Right**: Detected ships by WT in the COSMO-SkyMed X-band HH-polarization SpotLight image of Tokyo Bay, Japan.

### 7.2.4. Multi/Sub-Look-Based Methods [186–190]

The principle of multi/sub-look processing is described in Section 2.1 and Figure 5. For ship detection from the multi-look images, a small moving window is used in both the look-1 and look-2 sub-images (for the two-look example), and the cross-correlation coefficient is computed. As shown in Figure 5, the background clutter is uncorrelated between looks (it is a purpose of multi-look processing), but the images of ships have some correlation, provided that the images are within the moving window. Then, the ships can be detected in the coherence image, i.e., the image of correlation coefficient as high coherence against low coherence of the background clutter. The advantage of this method is that ships not clearly visible in SAR images can be detected. The disadvantage is that the detection may fail if a cruising ship is outside the window in the look-2 sub-image due to the inter-look time difference over which the ship moves. This method, however, is the basis of estimating the direction and cruising speed of moving ships as will be discussed later. Further, based on the principle of sub-look, the generalized likelihood ratio test (GLRT) [190,191] was developed for ship detection.

### 7.2.5. Polarimetric Analyses [30–32]

As mentioned in Section 7.2, HV-polarization is suitable for ship detection in comparison with co-polarization. Further, the approach using fully polarimetric and dual-polarization data has been proposed. In the model-based four-component scattering power decomposition, the total scattering power is decomposed into the surface, double-bounce, volume and helix scattering components. Note that the helix scattering process is the process by which the linearly polarized incident wave becomes a circularly polarized wave upon reflection by, for example, crossed wires. Considering the scattering processes from the sea surface and ships, the polarimetric power decomposition is suitable for ship detection. Figure 30 shows the color composite representation of the ALOS-PALSAR PLR data. The sea surface appears in blue, indicating the surface scattering, while the land areas

appear mainly in green, indicating the volume scattering from forests. The urban areas of structure aligned orthogonal to the range direction appear as double-bounce scattering as shown in red. Ships also appears in red and green due to the double-bounce and volume scatterings.

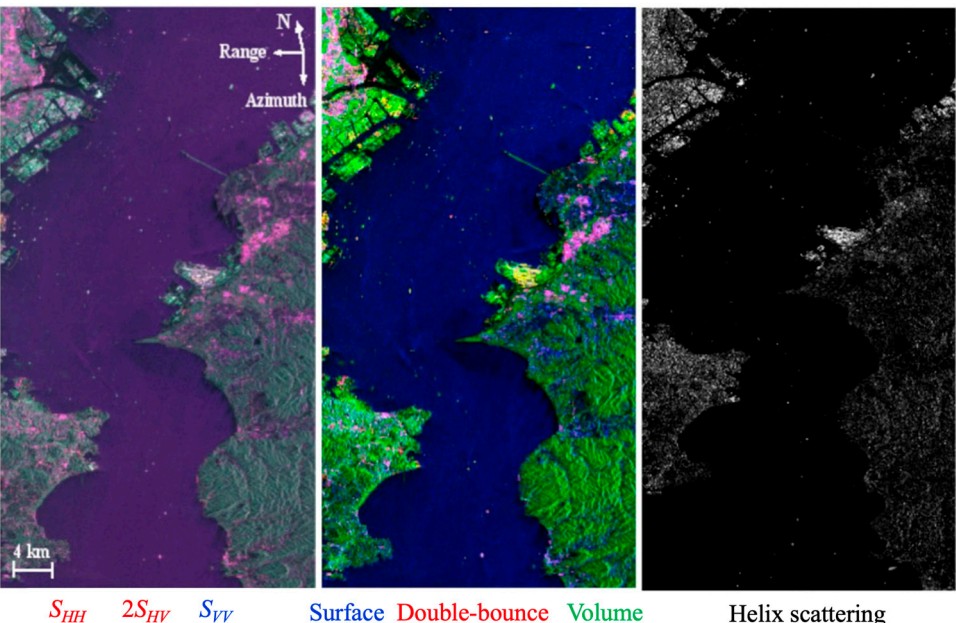

$S_{HH}$  $2S_{HV}$  $S_{VV}$      Surface  Double-bounce  Volume      Helix scattering

**Figure 30.** Color composite representation of the ALOS-PALSAR PLR images shown in Figure 27.

For the HH/VV dual-polarization SAR without HV-polarization, several approaches are possible, including the co-polarization entropy, correlation, and phase difference. The entropy is defined from the eigenvalue analysis as $H = -p_1 \log_2 p_1 - p_2 \log_2 p_2$, where $p_j$ is the probability for which the $j$th scattering process occurs. Thus, $H = 0$ means that there is only a single (surface) scattering process corresponding to the sea surface, while $H = 1$ means the backscattering of mixed (random) scattering processes from ships. The correlation coefficient is

$$\gamma = |\langle S_{HH} S^*_{VV} \rangle| / (\langle |S_{HH}|^2 \rangle \langle |S_{VV}|^2 \rangle)^{1/2}. \tag{4}$$

where $S_{HH}$ and $S_{VV}$ are the complex scattering amplitudes of corresponding polarizations. For the surface scattering, the inter-correlation increases as $\gamma \to 1$, and $\gamma \to 0$ for decreasing correlation due to multiple scattering from ships. The phase difference is $\delta\phi = \phi_{VV} - \phi_{HH}$, where $\delta\phi \to 0$ for the surface scattering, and $\delta\phi \to \pm\pi$ for the multiple scattering.

Figure 31 shows the TerraSAR-X dual-polarization SpotLight SAR image (HH-polarization) of Tokyo Bay, Japan (20 September 2012). The simultaneous observation was made by AIS and visual observation with a video camera. The entropy, inverse coherence $(1 - \gamma)$, and phase difference show much higher image contrast of ships in comparison with the intensity images. Although the number of test images is not large, the peak-to-background-image ratio was 0.83 for the entropy and 0.95 for the correlation, while those of the co-polarizations were 0.67. Although the number of samples is small, the results show the effectiveness of the entropy and inter-correlation methods.

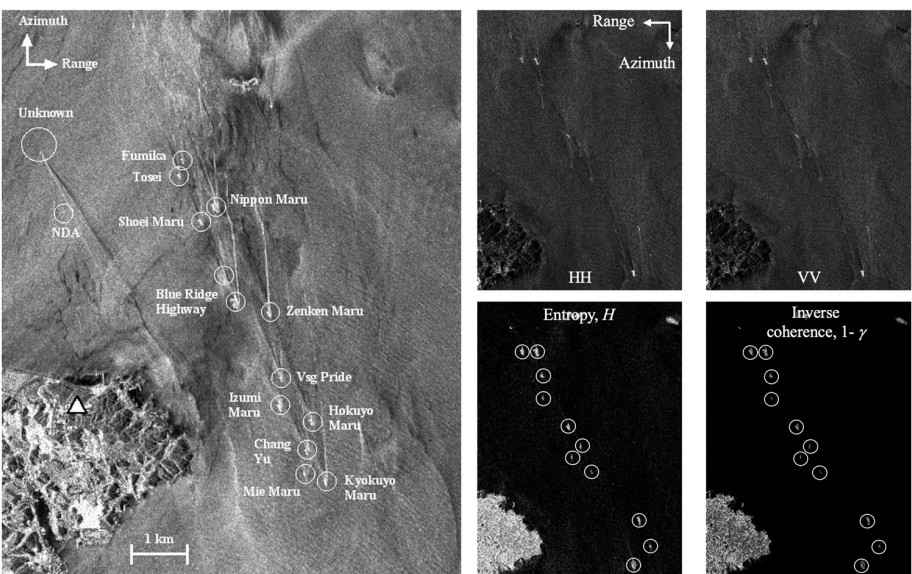

**Figure 31. Left**: TerraSAR-X HH-polarization SpotLight SAR image of Tokyo Bay, Japan, showing the detected and identified ships. **Right**: HH/VV-polarization images of the test area (upper) and the detection results (lower).

### 7.2.6. Standard Deviation Filter (SDF) [192]

In the SDF method, the standard deviation and mean are calculated in a moving window of a size large enough for statistical calculation. If a target is within the window, the standard deviation is larger than that of the window containing only clutter. In order to equalize the varying image amplitude along the range (decreasing amplitude with range), the standard deviation can be subtracted by the mean within the moving window. This approach requires the statistical computation in a large window, and the target size is larger than that of the pixel-based method such as the ATM.

### 7.2.7. AI-Based Method

In the recent years, many methods of ship detection using deep learning have been reported. For example, an automatic detection model was proposed based on the RetinaNet object detector with 86 scenes of Chinese Gaofen-3 SAR images at four resolutions (3 m, 5 m, 8 m, and 10 m) [45]. Two Gaofen-3 images and one Constellation of Small Satellite for Mediterranean Basin Observation (Cosmo-SkyMed) image were used to evaluate the robustness. The result showed the model can detect multi-scale ships efficiently with a high detection accuracy, with a mean detection precision over 96%.

A detection model utilizing the YOLO family was developed using two types of datasets: the SAR ship-detection dataset (SSDD) and diversified SAR ship-detection dataset (DSSDD) for training and testing [119]. The SSDD dataset is the first open dataset and a benchmark containing 1160 images and 2456 ships acquired by RADARSAT-2, TerraSAR-X and Sentinel-1 in various environments. The DSSDD is directly collected from these spaceborne SARs, with more diversity in ships and SAR resolutions. In the study, 50 SAR images with resolutions from 1 m to 5 m from the two datasets were used. The proposed YOLOv2 architecture reached an accuracy of 90.05% and 89.13% on the SSDD and DSSDD datasets, respectively.

Using the same SSDD dataset, a ship detection algorithm based on YOLO-v4-light was developed, showing a mean precision of 90.37% [120]. Further, a new improvement on the YOLOv3 based on SAR and optical images was proposed [121]. The SAR data were collected from the Gaofen-3 and Sentinel-1 SARs at four resolutions, and the optical data were extracted from SPOT imagery. By comparison with other models in the YOLOv3 family, the improved-YOLOv3 model showed the better average precision on both the optical (93.56%) and SAR (95.52%) datasets.

### 7.3. Ship Classification

As noted earlier, the information on cruising ships can be identified via AIS signals. Figure 32 shows the ALOS-2 PALSAR HH-polarization image (4 December 2016) showing the ships detected by the ATM with $c$ = 2.5, and identified with AIS in the northern waters of Taiwan [196]. As can be seen, the vast number of ships were without AIS signal. It is thus, necessary to classify these ships from SAR data alone. The progress of ship classification was slow compared with the ship detection mainly due to the spatial resolution not being fine enough, and also a small number of SAR images of the same ship for classification. In recent years, however, many spaceborne SARs have been launched and a number of airborne SARs have also been increased. Correspondingly, several new methods have been proposed for ship classification [197–204].

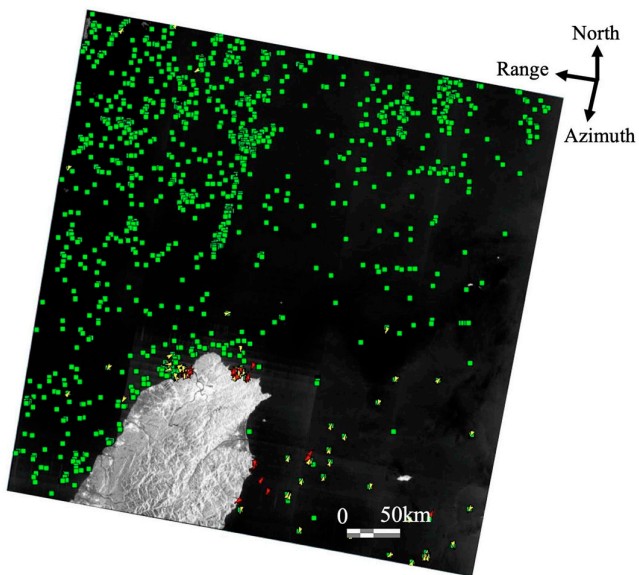

**Figure 32.** ALOS-2 PALSAR HH-polarization image (4 December 2016) showing the ships detected by the ATM with $c$ =2.5 and identified with AIS in the northern waters of Taiwan [196].

The flow of ship classification is illustrated in the left of Figure 33. From the input image, the geometrical and radiometric features such as the shape, size, and intensity in different parts are extracted. With these features, classification is made using the algorithms including feature-based template matching, pattern matching, multi-channel analyses and machine learning.

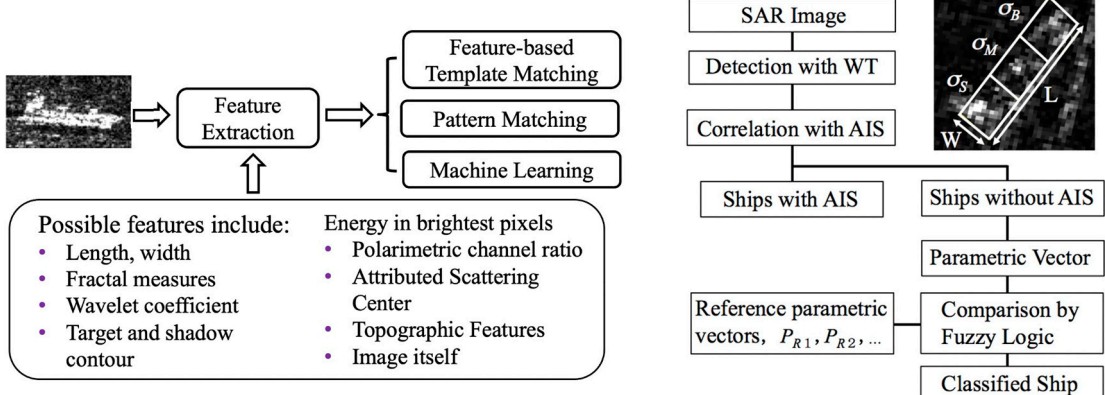

**Figure 33.** Left: Flowchart for ship classification. Right: Flowchart for ship detection and classification by the GMV Spain.

The feature-based template matching using the parametric vectors was developed by several institutes, for example, the GMV Spain [197,198], the National University of Defense Technology, China [199], and Lockheed Martin Canada (LMC) [200]. The flowchart on the right of Figure 33 shows the system developed by the GMV Spain. Ship candidates are first detected using the WT, and matching with AIS data is carried out. The parametric vector is then extracted from the images of ships without AIS and those not matched with AIS. The parametric vector consists of the image length $L_{Ship}$, width $W_{Ship}$, and the intensities $\sigma_B$, $\sigma_M$, and $\sigma_S$ in the bow, middle, and stern sections, respectively. The reference parametric vector is computed prior to the comparison using the simulated or real SAR images of different ship categories. Finally, the classification is made by comparison of the extracted and simulated parametric vectors using fuzzy logic. The accuracy was tested using the TerraSAR-X image of Tokyo Bay shown in Figure 31 and the Mediterranean Sea, acquired on 4 December 2012 [201]. Unfortunately, only three ships were visible in the latter image because of very strong wind. A part of the results is shown in Figure 34. For the reference images, ships of seven classes were simulated: passenger, tanker, cargo/bulk container, vehicle carrier, medium cargo, and medium and small fishing boats. Although the number of ships was 18, the classification accuracy was 88%. These results were slightly better than the previous results with an accuracy of ~70% using ENVISAT-ASAR data, mainly because of the higher resolution (3 m) than that of the ASAR (30 m) [198].

(number): true values  Tokyo Bay Data

| Name | Type | L | W | $\sigma_S$ | $\sigma_M$ | $\sigma_B$ | Classified as | |
|---|---|---|---|---|---|---|---|---|
| Tosei | Container | 99.5 (85) | 7.2 (14) | 43.1 | 41.3 | 42.9 | Medium Cargo | |
| Fumika | Cargo | 119.2 (73) | 10.2 (12) | 43.1 | 39.8 | 43.6 | Medium Cargo | |
| Shoei Maru | Oil/Chemical Tanker | 112.4 (105) | 9.5 (16) | 43.1 | 41.9 | 40.1 | Tanker | |
| Nippon Maru | Passenger | 217.2 (166) | 22.5 (25) | 40.6 | 42.4 | 40.4 | Container | |
| Blue Ridge Highway | Ro-ro/ Passenger | 150.0 (180) | 21.4 (31) | 36.3 | 37.7 | 32.4 | Medium Cargo | |
| Zenken Maru | Waste Disp. Vessel | 87.4 (98) | 10.9 (18) | 52.3 | 51.0 | 50.9 | Medium Cargo | |
| Vsg Pride | Cargo | 95.2 (97) | 11.0 (17) | 46.2 | 47.8 | 42.9 | Medium Cargo | |

**Figure 34.** A part of the classification results by the feature-based template matching of the TerraSAR-X data shown in Figure 31. The length (L) and width (W) in the brackets are the true values of AIS data [201].

The method of pattern matching needs a large number of simulated or real SAR images of a ship from different look angles and incidence angles. There have been studies on the classification of military vehicles by real aperture radars, but no study, to the authors' knowledge, has been reported on the pattern matching with SAR images. The multi-channel analyses are those using polarimetric and InSAR images for template matching.

AI has recently been applied to ship classification as well as various other fields. For example, a deep learning network was developed for ship classification and characterization using Sentinel-1 images and matched AIS data [202]. The model was a multi-task neural network for three specific tasks, namely, ship detection, classification and ship length estimation. For classification, a four-class database (tanker, cargo, fishing, passenger) was

used for comparison with other approaches, i.e., a simple mult-layer perceptron (MLP) and R-CNN, and a five-class database (four classes and tug boat) was used for the accuracy estimation by the proposed model. The networks were trained using 16,000 images and 4000 different images for validation. The results showed that the overall classification accuracy by MLP was 25%, with a length estimation mean error of $-7.5$ m $\pm$ 228 m. R-CNN performed rather well, with the overall classification accuracy of 89.29%, while, the overall classification accuracy of 97% and the mean length error of 4.65 m $\pm$ 8.55 m was achieved by the proposed model. As to the ship size estimation, a deep learning model named SSENet was developed, employing a single shot multibox detector-based model with a rotatable bounding box. The training and testing dataset were 1500 and 390 ships. The result showed the mean absolute errors are under 0.8 pixels; in other words, the length and width are 7.88 and 2.23 m, respectively.

As an application of semantic segmentation, the Gaofen-3 (GF-3) data were used to classify eight types of targets (boat, cargo ship, container ship, tanker ship, cage, iron tower, platform, and windmill) [203]. For classification at a patch level, a novel CNN model with six convolutional layers was designed, and a single shot multi-box detector with a multi-resolution input (MR-SSD) was developed for detection. A total of 111 GF-3 fully-polarimetric SAR images were selected over the offshore areas of Eastern Asia, Western Asia, Western Europe, and Northern Africa from December 2016 to May 2018. The resolution was from 0.5 m to 5 m, and 2522 training data were used with 688 test data. The proposed method showed good category classification results except for platforms, with the mean accuracy achieving 95%.

Similar results using CNN were reported for ship detection and classification with TerraSAR-X StripMap high-resolution (~3 m) images at different incidence angles (ranging from 20° to 45°) and in different polarizations (73% for HH, 27% for VV) [204]. The category was composed of five maritime classes of cargo, tanker, windmill, platform, and harbor structure. A total of 683 target data were extracted (68% of ships, 32% of other structures), and the proposed model showed a classification performance with a total average F-score of 0.94.

## 8. Wind

There are several approaches to the wind measurements using SAR. The scatterometer-based approach is based on the relation between the NRCS and the empirical geophysical model function (GMF), known as the CMOD4, first developed for the C-band VV-polarization microwave wind scatterometer onboard the ERS-1 satellite of ESA [205]. Since then, different versions in the CMOD family have been developed at different frequencies, ranging from Ku-band to L-band [206–208]. The backscatter RCS from the sea surface depends on the incidence angle $\theta_i$, wind speed, and direction relative to the azimuth look angle. The GMF takes a general form:

$$\sigma_0{}^{\mathrm{PP}} = B_0 \, (U_{10}, \theta_i)[1 + B_1(U_{10}, \theta_i) \cos\phi + B_2 \, (U_{10}, \theta_i) \cos(2\phi)]^c \tag{5}$$

where $\sigma_0{}^{\mathrm{PP}}$ is the NRCS in PP-polarization, $B_j$ ($j$ = 0, 1, 2) coefficients are empirically determined by matching the scatterometer data with reference wind, the $U_{10}$ is the wind speed at 10 m above the sea surface, $c$ is a constant, and $\phi$ is the wind azimuth angle, which should not be confused with the polarimetric phase. The reference winds can be measured by buoy or the numerical weather prediction (NWP) model.

Figure 35 shows an example of a GMF. The scatterometer measures NRCS at different wind azimuth angles with multiple antennas or by scanning with a rotating antenna. For example, the Ku-band NSCAT onboard the Advanced Earth Observation Satellite (ADEOS) used six fan beam antennas, and the SeaWinds scatterometer onboard QuikSCAT uses a rotating antenna [209]. The recent C-band ASCAT on the meteorological operational (MetOp) platform also uses six antennas [207]. The wind speed and direction can then be measured by fitting the NRCS at different azimuth angles with the GMF. The spatial resolution of scatterometers is over 12 km, and thus it is difficult to measure the wind

vector near coastal waters. The wind measurements using fine-resolution SAR can fill the gap in the wind scatterometer data.

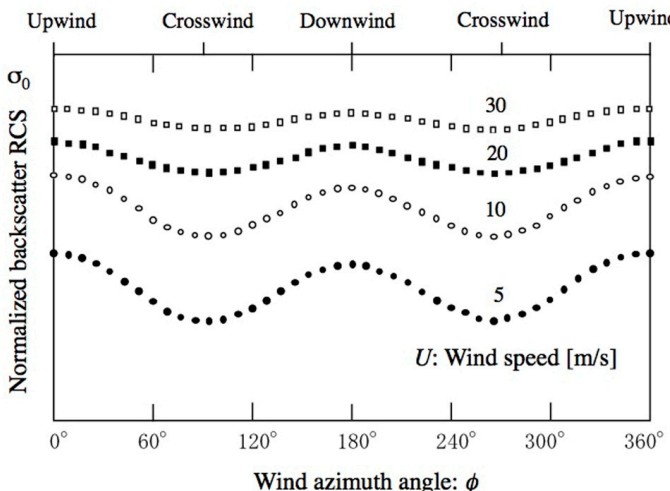

**Figure 35.** A form of CMOD GMF showing the NRCS for different wind azimuth angles of wind and wind speeds.

Unlike scatterometers that use multiple beams at different azimuth look angles, only the line-of-sight (range) velocity component can be measured by SAR with a single beam. In general, wind directions are estimated by combining other sources such as the image spectra and wind streaks in SAR images [208,210,211].

The CMOD GMF was developed with VV-polarization data. To convert GMF from VV-polarization to HH-polarization GMF, the co-polarization ratio model

$$\sigma_0^{HH}/\sigma_0^{VV} = (1 + a \tan^2\theta_i)^2/(1 + b \tan^2\theta_i)^2 \qquad (6)$$

was proposed, where $a$ and $b$ are the empirical constant depending on the scattering models and incidence angles [208,210,212]. Note that an empirical model for the wind speed measurements by the C-band cross-polarization data, $\sigma_0^{VH} = 0.592 \, U_{10} - 35.6$, was also developed [213].

Figure 36 shows an example of the wind direction estimation using the image spectra [210]. The right image shows the ERS-1 C-band VV-polarization SAR image off the northwest coast of the Shetland Islands (1 December 1992). The solid lines show the wind direction estimated by the method of the image spectrum showing in the right image. In the right image, the wavelength of ocean waves can be estimated by the two spectral peaks, indicating the 180° directional ambiguity. The weighted cross-spectra described in Section 2.2 can resolve this ambiguity. Nevertheless, the ellipsoid shows the spectral distribution of the wind streaks in the direction orthogonal to the ellipsoid. Again, there is a 180° directional ambiguity, but it can be resolved if there is shadowing over the wind field by, for example, the coastal topography [210].

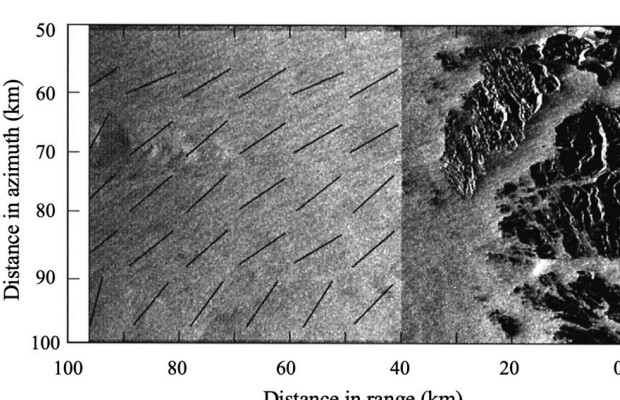 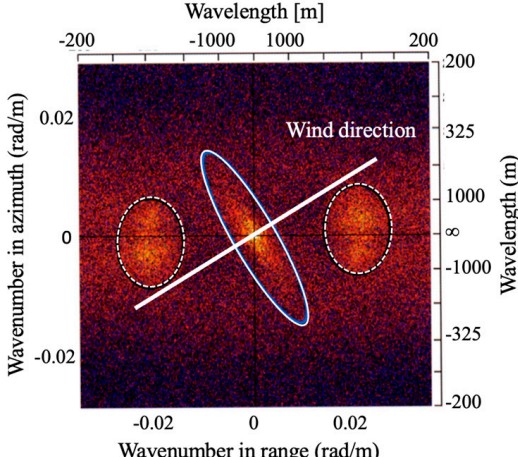

**Figure 36.** **Left**: ERS-1 SAR image off the northwest coast of the Shetland Islands. The solid lines are the wind direction with 180° directional ambiguity estimated from the image spectrum shown in the right. **Right**: Image spectrum computed by Fourier transform of a part of the left SAR image showing the wavelength (~300 m) of ocean waves in both the directions from the center. The ellipsoid shows the direction orthogonal to the wind streaks [210].

With increasing interest in AI applications, neural network methods were used to retrieve sea surface wind speed from HH-polarized Sentinel-1 SAR images [214]. The polarization ratio models combined with the CMOD5.N were applied. A total of 130 Sentinel-1 data sets in EW and IW modes were collected from October 2014 to December 2018. The buoy data were also matched-up with the SAR data. Compared to the buoy data, the bias and root mean square error of the wind speed retrieved by the neural network model were 0.10 m/s and 1.38 m/s, respectively.

Another approach was proposed based on the cutoff wavelength of ocean wave images [215,216]. It was noted in Figure 7 that the resolution in the azimuth direction is degraded and the PDSF is smeared due to the random motion of scatterers (Bragg waves). The images of ocean waves under different wind speeds results in different amount of smearing. Consequently, there is an upper limit of measurable wavelength called the cut-off wavelength $\lambda_c$ or cut-off wavenumber $k_c = 2\pi/\lambda_c$. The empirical relation, $\lambda_c = 23.4\ U_{10} + 70.0$ [m], was proposed using the relation of the azimuth cut-off wavelength and wind speed [215]. This method was tested using the ERS-2 SAR wave mode data and compared with the collocated scatterometer SCAT data. Although the correlation between the measured image smearing and the SCAT data was reasonable, the method does not seem to be comparable with the CMOD algorithms [216].

It should be emphasized that the ocean wind field measurements by SAR are of current practical interest and also for near operational use such as the Italian Space Agency-funded APPLICAVEMARS project [217].

## 9. Other Oceanic Phenomena

### 9.1. Aquaculture

With the rapid increase in aquaculture during the last several decades, monitoring of coastal aquaculture of various species and fisheries is an important issue for management, conservation, development and restoration purposes [218]. An early observation of fishing activity by SAR was reported in 1992 during the Halieutis Radar Experimentation Mediterranean Sea (HAREM) conducted in 1989, employing the C-band VV-polarization E-SAR by the German Aerospace Agency (DLR) [219]. The data revealed the purse seine fishing activity, floats and nets above the sea surface as bright features. The previous experiment with the French VARAN-S X-band SAR also showed oyster tables, fish traps and nets above the sea surface [219].

Figure 37 shows the RADARSAT-1 SAR images of large fish cargoes on the left and fish traps on the right. Again, these structures are above the sea surface, giving rise to strong radar backscatter through the double-bounce scattering between the sea surface and the cargoes and fish traps. The detection of such bright images against the dark background can be made in a similar manner with target detection, such as ship detection.

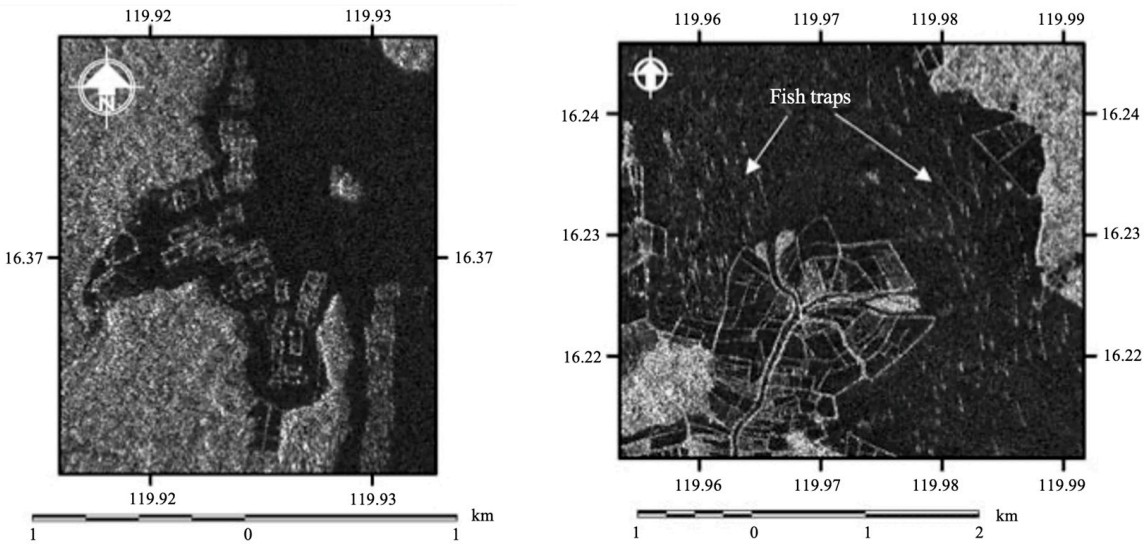

**Figure 37.** RADASAT-SAR images of fish cargoes on the left, and fish traps on the right as bright features against dark background of the sea surface (4 February 2001) [220].

As a practical application, continental-scale mapping was carried out to detect the pond aquaculture for the entire coastal zone of Asia using 25,000 Sentinel-1 SAR time-series images and Sentinel-2 multispectral images [221]. Note that pond aquaculture here is the farming of fish, shrimp, etc. for food in a controlled environment, such as enclosed and semi-enclosed nets. A histogram-based thresholding method was used for the automatic detection and extraction of aquaculture areas on a single-pond basis. Over 3.4 million ponds in an area of over 2 million hectare were detected and mapped along the shoreline of South Asia, Southeast and East Asia. A similar study on a local basis using 113 Sentinel-1 SAR scenes and 328 Sentinel-2 MSI scenes was reported to monitor the aquaculture in Palawan, the Philippines from 2016 to 2019, where significant change of aquaculture spatial distribution was observed [222].

Methods based on deep learning have recently been applied to the extraction of aquaculture areas [223–225]. The principal approach is essentially the same as that of oil slick and ship detection, using a substantial amount of data for training, followed by applying the test data for the algorithm evaluation.

The above-mentioned methods are for the detection and classification of marine cultivation targets above the sea surface. A study was conducted to detect underwater cultivation fields using polarimetric analyses [33,226]. Figure 38 shows a part of the ALOPS-PALSAR PLR images around the Futtsu peninsula on the right (white rectangular box in Figure 27); from left to right, the HH-polarization amplitude image, entropy image, and a photograph of the test site are shown. Laver (*Porphyra*) is edible algae rich in protein, dietary fiber, vitamins, and a source of mineral extract for the food and medical industries. Laver cultivation is an important marine industry in Asia, along with oyster and fish farming. After seeding, the cultivation nets are placed at the depth of 10–20 cm. As in the figure, the image contrast of the surface above the cultivation nets is low (0.25, 0.12, and 0.38 for HH, HV, and VV, respectively) in comparison with the contrast of the entropy image (0.56). The in situ observation in the right image shows that the surface above the underwater nets is smooth compared with the surrounding deep water, i.e., the area is

effectively shallow water. As a result, the backscattered signal is within the random system noise level, yielding high entropy, similar to the polarimetric analysis of oil spills [98,117].

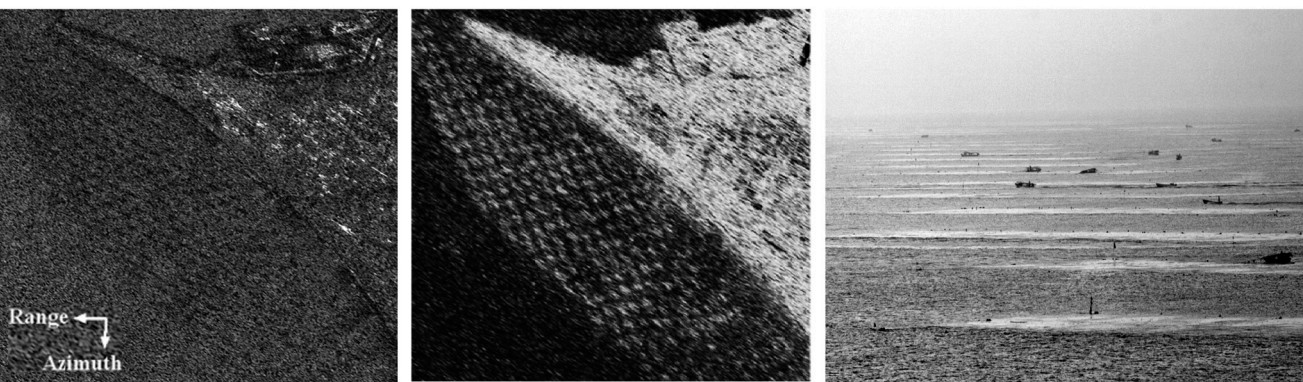

**Figure 38.** **Left**: ALOS-PALSARHH-polarization image of the laver cultivation area in Tokyo Bay, Japan. **Center**: Entropy image. **Right**: Photograph of the test site.

At present, the methodology of detection and extraction of aquaculture is fairly well established and these algorithms are applied to routine observation in the local, regional and global scales, as for those reported in the examples described above [221,222].

*9.2. Sea Ice*

Approximately 15 percent of the global oceans are covered by sea ice during part of the year, in particular, in the polar regions. With increasing global warming, the spatial and temporal changes of sea ice are an indication of the changes in global climate [227]. There are more than a dozen operational ice monitoring services to support climate change studies, and ship navigation in ice-covered waters in near-real time. In addition to measurements taken in situ and by airborne sensors, spaceborne SAR and optical data are widely used.

The imaging process of sea ice by SAR is illustrated in Figure 39. The single-bounce surface scattering is dominant from the open water. The backscattering increases from newly-borne ice, and further increases from the rough first-year ice, containing much salt. The double-bounce scattering also occurs between the ice ridge and water surface. Multi-year ice contains less salt and the surface becomes smoother than the first-year ice. The radar backscatter is due mainly to the surface scattering and double-bounce scattering at the ice ridges as well as volume scattering from the ice interior. The volume scattering increased with increasing radar wavelengths, i.e., L/P-band.

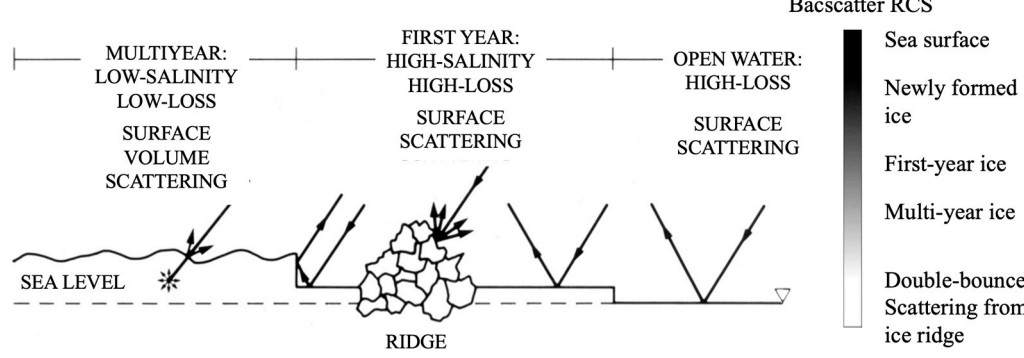

**Figure 39.** Illustrating the radar backscatter on sea ice and open water [5].

Figure 40 shows the total power and composite color images of the Beaufort Sea acquired by the NASA/JPL airborne AIRSAR (11 February 1988), where dependence of sea ice images on different frequency bands can be seen [34]. Since then, there have been many studies reported for sea ice detection and classification on different frequencies and

polarizations over several waters, e.g., [228,229]. As a result, the C-band HH-polarization SAR is considered to be suitable in comparison with the other frequencies and polarizations if only a single polarization is used. If data are available, polarimetric analyses are better for sea ice discrimination [228], and also the floe size distribution and temporal changes by both optical and SAR data [230].

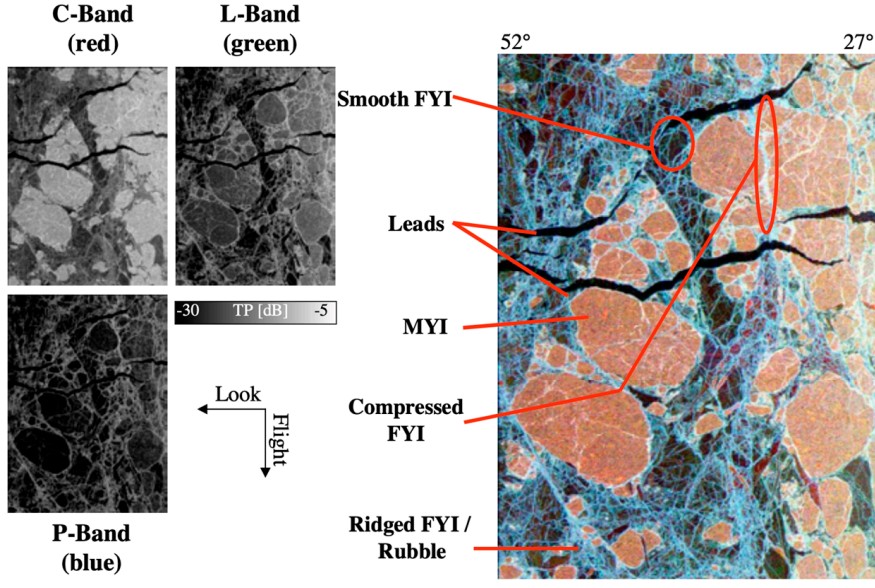

**Figure 40.** The total power (**left**) and composite color images (**right**) of the Beaufort Sea acquired by the NASA/JPL airborne AIRSAR (11/02/1988) [34].

As for the other oceanic phenomena, AI has been applied to sea ice classification in recent years [231–233]. The classification of the sea ice type was reported using the 18 C-band SAR GF-3 quad-polarization StripMap data over the Arctic Ocean in 2017 [231]. Floe ice, brash ice between floes, and open water (ice-free area) were classified based on a mini sea ice residual convolutional network called as MSI-ResNet. The combination of VV and VH-polarizations resulted in a modest precision improvement for brash ice and open water together with a slight overestimation for floe ice. With the VV, VH and HH-polarization data as the inputs, the accuracy reached 95.12%, 93.42%, and 95.17% for floe ice, brash ice and open water, respectively.

The Gaofen-3 satellite in the dual-polarization (VV, VH) fine strip II (FSII) mode in the Arctic area in winter was also used for sea ice classification [232]. The western Arctic Ocean from January to February 2020 was selected. The classification was made for the sea ice into four categories, namely new ice, thin first-year ice, thick first-year ice, and old ice, by referring to the ice maps provided by the Canadian Ice Service (CIS). The deep learning model called Multiscale MobileNetV3 was developed based on MobileNet (MSMN) as the backbone network with input samples of different sizes, and combining the backbone network with multiscale feature fusion methods. As a result of testing with SAR sea ice images, the classification accuracy reached over 95%.

A study was conducted for the automatic estimation of sea ice concentration by the fully convolutional network using the Sentinel-1 dual-polarized C-band SAR images [233]. A database was generated with 1320 dual-polarized S-1scenes with ice charts produced by MET Norway. A comparative benchmark was carried out with Ocean and Sea Ice Satellite Application Facility (OSISAF) and MET Norway sea ice concentration products. The result showed an overall accuracy of 78.2% for six-class classification.

## 10. Conclusions

A summary was presented on the SAR imagery of oceanic phenomena. Since the first civilian spaceborne SEASAT-SAR for ocean applications as the main purpose, many spaceborne SARs have been launched, and considerable effort has been made to improve the algorithms and systems. Currently, the technology of SARs is at a mature stage, and routinely operating for practical use. Among many oceanic phenomena, the present article focused on the ocean surface and internal waves, oil slicks, ocean currents, bathymetry, ship detection and classification, wind, aquaculture, and sea ice. The methodology is based on the conventional use of image intensity, interferometry, polarimetry and artificial intelligence. Examples and references were provided where necessary; for further examples and information, please refer to the references therein.

**Author Contributions:** Conceptualization, K.O. and T.Y.; original draft preparation, K.O.; writing—review and editing, K.O. and T.Y. All authors have read and agreed to the published version of the manuscript.

**Funding:** This research received no external funding.

**Data Availability Statement:** Not applicable.

**Acknowledgments:** ALOS data are the property of the Japan Aerospace Exploration Agency (JAXA). ERS-1/2, ENVISAT, Sentinel and COSMO-SkyMed data are the property of the European Space Agency (ESA). RADARSAT and TerraSAR-X data are the properties of the Canadian Space Agency (CSA) and German Aerospace Agency (DLR) respectively. Pi-SAR data are the property of the National Institute of Information and Communications Technology (NICT) of Japan.

**Conflicts of Interest:** The authors declare no conflict of interest.

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
