# Peer review of "On the Interpretation of Synthetic Aperture Radar Images of Oceanic Phenomena: Past and Present"

_remotesensing, doi:10.3390/rs15051329_

Round 1
Reviewer 1 Report
The review presents the different cases of the analyses of the SAR-produced images of various sea/ocean phenomena and objects. A huge amount of references, clear, thorough introduction and descriptions of the subsequent cases make this review a very attractive source for further research in more specific areas. A few details require attention:
32, 35-38, 87-88, 96, 133-134, 221, 235, 368, 388, 392, 563, 569, 624, 679, 751-753, 855, 889, 986, 1072, 1326, 1328, 1329, 1331, 1377, 1407, 1423, 1517, 1713, 1775: the reference indication could be organized in a shorter form, i. e. [1-4], [4, 6-10], [11-16] etc.;
100: 'including those observed in Figure 1' - does this refer to the waves in Figure 1?
181: What is the unit of v_SR?
231: Should it be 'multiplied'?
233/ Figure 5: Does the asterisk indicate that the complex conjugate is taken from S2 only?
Figure 12: The time variable (570-571) is not in italics, as previously introduced;
618: A full stop instead of a comma;
1018, 1021: Is this Q the wind drift factor? (Italic in the previous mentions);
1262: 'the' instead of 'he'; this large equation (Omega^2 = ...) could gain in legibility if written in a separated form, not within the main text (same for eqs. in 1360, 1363, 1454, 1458, 1612-1613, 1645);
1362: Should the '<z>' be in italics?
1540: red colour.
Author Response
Reply to the Reviewer 1:
We thank you for the valuable comments to improve the manuscript. We revised the manuscript according to your comments. We also made minor corrections.
Comments and Suggestions for Author: The review presents the different cases of the analyses of the SAR-produced images of various sea/ocean phenomena and objects. A huge amount of references, clear, thorough introduction and descriptions of the subsequent cases make this review a very attractive source for further research in more specific areas. A few details require attention:
32, 35-38, 87-88, 96, 133-134, 221, 235, 368, 388, 392, 563, 569, 624, 679, 751-753, 855, 889, 986, 1072, 1326, 1328, 1329, 1331, 1377, 1407, 1423, 1517, 1713, 1775: the reference indication could be organized in a shorter form, i.e. [1-4], [4, 6-10], [11-16] etc.;
Reply: The reference indications are in shorter forms.
100: 'including those observed in Figure 1' - does this refer to the waves in Figure 1?
Reply: “those observed in Figure 1” is replaced by “surface waves”.
181: What is the unit of v_SR?
Reply [m/s]
231: Should it be 'multiplied'?
Reply: Yes, it is “multiplied”.
233/ Figure 5: Does the asterisk indicate that the complex conjugate is taken from S2 only?
Reply: Yes, it is only S2. Also S1 × S2* (230)
Figure 12: The time variable (570-571) is not in italics, as previously introduced;
Reply: The time variable t is in italics throughout the text.. Figure 12 is also updated.
618: A full stop instead of a comma;
Reply: Changed.
1018, 1021: Is this Q the wind drift factor? (Italic in the previous mentions);
Reply: Yes. Changed to “Q” in italics.
1262: 'the' instead of 'he'; this large equation (Omega^2 = ...) could gain in legibility if written in a separated form, not within the main text (same for eqs. in 1360, 1363, 1454, 1458, 1612-1613, 1645);
Reply: Changed as ”the”. Equations (1)-(6) are in the separate form, but the others are kept within the lines.
1362: Should the '<z>' be in italics?
Reply: Yes. Changed.
1540: red colour.
Rely: Done.
Further corrections mainly for avoiding the repetition of same expressions already previously given.
15: “naritime” to “maritime”
89: “artificial intelligence (AI)” to “(AI)”
101: Deleted “and snow”
274: Deleted “Deep Neural Network”
282: Deleted “Convolution Neural Network”
512: Added the reference [24]
827-828: Deleted “(Automatic Identification)”
885-888: Deleted “artificial intelligence”, “support vector machine”, “neural network” and “convolution neural. network”
893-897: Added the references
960: Added the reference [234]
1483: “several” to “many”
1569-1570: Replaced “machine learning” by “AI”. Deleted “such as pattern recognition, medical diagnosis, search
engines and robotics”
1603: Replaced “f1” by “F”
1659: Deleted “conventional retrieval and”
1774-1778: Replaced “machine learning” by “AI”, deleted “Gaofen-3”,“(QPS)”, “(FI)”, “(BI)” and “OW”
1780-1782: Replaced “BI” by “brash ice”, “OW” by “open water”, “FI” by “floe ice”, “FI”
1786-1787: Deleted “(NI)”, “(tI)”, “(TI)”, and “(OI)”
1792-1793: Deleted “(SIC)”, “(FCN)” and “(SI)”
1796: Replace “ocean surface and SIC” by “sea ice concentration”
1804: Replace “Ocean surface and internal waves” by “ocean surface waves, internal waves”
Reviewer 2 Report
This is very well organized article with the extensive overview of the past and present papers. In my opinion this article could be published with rather small grammatical changes and probably increased quality of some figures (i.e., labels on the fig. 16d are not readable)
Author Response
Reply to the Reviewer 2
Comments and Suggestions for Author: This is very well organized article with the extensive overview of the past and present papers. In my opinion this article could be published with rather small grammatical changes and probably increased quality of some figures (i.e., labels on the fig. 16d are not readable)
Reply: We thank you for your valuable comment. The labels in Figure 16d are now readable.
Further corrections mainly for avoiding the repetition of same expressions already previously given.
15: “naritime” to “maritime”
89: “artificial intelligence (AI)” to “(AI)”
101: Deleted “and snow”
274: Deleted “Deep Neural Network”
282: Deleted “Convolution Neural Network”
512: Added the reference [24]
827-828: Deleted “(Automatic Identification)”
885-888: Deleted “artificial intelligence”, “support vector machine”, “neural network” and “convolution neural. network”
893-897: Added the references
960: Added the reference [234]
1483: “several” to “many”
1569-1570: Replaced “machine learning” by “AI”. Deleted “such as pattern recognition, medical diagnosis, search
engines and robotics”
1603: Replaced “f1” by “F”
1659: Deleted “conventional retrieval and”
1774-1778: Replaced “machine learning” by “AI”, deleted “Gaofen-3”,“(QPS)”, “(FI)”, “(BI)” and “OW”
1780-1782: Replaced “BI” by “brash ice”, “OW” by “open water”, “FI” by “floe ice”, “FI”
1786-1787: Deleted “(NI)”, “(tI)”, “(TI)”, and “(OI)”
1792-1793: Deleted “(SIC)”, “(FCN)” and “(SI)”
1796: Replace “ocean surface and SIC” by “sea ice concentration”
1804: Replace “Ocean surface and internal waves” by “ocean surface waves, internal waves”
Reviewer 3 Report
This is an interesting and rather complete review f the different uses of SAR data over the ocean, both for a variety of scientific or monitoring purposes. The explanation of what is the principle in the data return with the different polarisations and different instruments used is very valuable. The structure of the paper seems fine to me, and my comments are mainly minor. However, as I am not an expert in Deep learning, I got a bit distracted, as its use and comments on the methods used and their performance came in quite a few places, and not in a specific section. However, I understand that it is difficult to proceed otherwise.
My minor comments are below:
Two first sentences of abstract unnecessary
l. 21: instead of ‘summary’, use ‘review’
Use of ‘The’ throughout the paper not always appropriate (‘The’ is a definite article). For example on l. 34, no ‘The’ in front of ‘new technologies’ or *on l. 37 in front of ‘new applications’. I suggest that the authors check the use of the ‘definite article throughout the paper (this part of the sentence should read ‘, as well as recent applications of artificial intelligence (AI) methods, such as deep learning…’). Or on l. 82, replace ‘the antenna separation’, but ‘an antenna separation…’. I will not edit the paper, but parts of the paper are difficult to read as is.
Also, quite a few typos (for example on lines 315-316, or just below on top left panel of figure 7)
Starting at line 71, use of many acronyms of methods or approaches: ATI SAR, PolSAR, PolInSAR for methods approaches, not always fully described (when they first appear, but to a large extent this is done later).
L. 127, reminds reader of the launch date of Seasat (and mission duration)
l. 394-396: I don’t understand the sentence which needs to be rewritten (as is , it does not make sense ‘effects of what on what?)
Comments on figure 11, l. 528: method a) yields a good agreement, but notice that the observed speed is larger by 20% than the theoretical one.
l. 612: hard to understand.
l. 1125: ATI SAR (and not AIT SAR)
l. 1359: I am not sure that it should be ‘clutter’ (as used in ‘the clutter of a distribution’, but I am not familiar with this field; later, what is used is ‘clutter distribution’ as on line 1372)
l. 1611: strange sentence.
l. 1677-1679. I understand the interest for specific dedicated studies, but is it also for near operational use?
l. 1703: I am not familiar with the expression ‘pond aquaculture’ to describe semi-enclosed fish farms/aquaculture structures/cultivation nets, but I guess from the title of paper 217 that it is now the ‘official’ term (maybe worth pointing out what is included in this category)
Author Response
Reply to the Reviewer 3
We thank you for valuable comments to improve the manuscript. We have made corrections according to your comments as follows.
Comments and Suggestions for Author:
This is an interesting and rather complete review f the different uses of SAR data over the ocean, both for a variety of scientific or monitoring purposes. The explanation of what is the principle in the data return with the different polarisations and different instruments used is very valuable. The structure of the paper seems fine to me, and my comments are mainly minor. However, as I am not an expert in Deep learning, I got a bit distracted, as its use and comments on the methods used and their performance came in quite a few places, and not in a specific section. However, I understand that it is difficult to proceed otherwise.
My minor comments are below:
Two first sentences of abstract unnecessary
Reply: The first two sentences in the abstract were removed and placed in the first lines of Introduction.
- 21: instead of ‘summary’, use ‘review’
Reply: Changed as “review”.
Use of ‘The’ throughout the paper not always appropriate (‘The’ is a definite article). For example on l. 34, no ‘The’ in front of ‘new technologies’ or *on l. 37 in front of ‘new applications’. I suggest that the authors check the use of the ‘definite article throughout the paper (this part of the sentence should read ‘, as well as recent applications of artificial intelligence (AI) methods, such as deep learning…’). Or on l. 82, replace ‘the antenna separation’, but ‘an antenna separation…’. I will not edit the paper, but parts of the paper are difficult to read as is.
Reply: We thought “interferometric SAR (InSAR) and polarimetric SAR (PolSAR)” were definite as “The new technologies”. Nevertheless, we made changes according to your comments. Also replaced “the AI” by “AI”, “the CNN” by “CNN”, and others.
Also, quite a few typos (for example on lines 315-316, or just below on top left panel of figure 7)
Reply: Thank you for pointing out the typos. We made corrections.
Starting at line 71, use of many acronyms of methods or approaches: ATI SAR, PolSAR, PolInSAR for methods approaches, not always fully described (when they first appear, but to a large extent this is done later).
Reply: Brief descriptions were given in the first paragraph of Introduction.
- 127, reminds reader of the launch date of Seasat (and mission duration)
Reply: Added “in 1978”
- 394-396: I don’t understand the sentence which needs to be rewritten (as is , it does not make sense ‘effects of what on what?)
Reply: The effect of IWs on and EL Niño -Southern Oscillation (ENSO) is suggested by Matthews et. al. [70]. We added a short explanation as follows.
The mechanism is based on the strong seasonal energy (heat) transport by IWs between the Java Sea and Indian Ocean through the Lombok Strait where the both monsoon- and El Niño-induced variability is strong. As such, the throughflow likely varies on interannual time scales to reflect an ENSO influence by reduced and enhanced transports during EL Niño years [70].
Reply: “the CNN” AI without “the”
Reply: Corrected.
Comments on figure 11, l. 528: method a) yields a good agreement, but notice that the observed speed is larger by 20% than the theoretical one.
Reply: We are not sure on the reason for the discrepancy.
- 612: hard to understand.
Reply: Corrected to “but the X-band image modulation was underestimated by amount of an order of magnitude the observed image modulation.”
- 1125: ATI SAR (and not AIT SAR)
Reply: Corrected.
- 1359: I am not sure that it should be ‘clutter’ (as used in ‘the clutter of a distribution’, but I am not familiar with this field; later, what is used is ‘clutter distribution’ as on line 1372)
Reply: We added “(noise)” as “ the clutter (noise) model” (Line 1353). Changed as “the distribution of clutter”
- 1611: strange sentence.
Reply: Corrected.
- 1677-1679. I understand the interest for specific dedicated studies, but is it also for near operational use?
Reply: We added “and also for near operational use”
- 1703: I am not familiar with the expression ‘pond aquaculture’ to describe semi-enclosed fish farms/aquaculture structures/cultivation nets, but I guess from the title of paper 217 that it is now the ‘official’ term (maybe worth pointing out what is included in this category)
Reply: We added the following sentence. Note that pond aquaculture here is the farming of fish, shrimp, etc. for food in a controlled environment such as enclosed and semi-enclosed nets.
Further corrections mainly for avoiding the repetition of same expressions already previously given.
15: “naritime” to “maritime”
89: “artificial intelligence (AI)” to “(AI)”
101: Deleted “and snow”
274: Deleted “Deep Neural Network”
282: Deleted “Convolution Neural Network”
512: Added the reference [24]
827-828: Deleted “(Automatic Identification)”
885-888: Deleted “artificial intelligence”, “support vector machine”, “neural network” and “convolution neural. network”
893-897: Added the references
960: Added the reference [234]
1483: “several” to “many”
1569-1570: Replaced “machine learning” by “AI”. Deleted “such as pattern recognition, medical diagnosis, search
engines and robotics”
1603: Replaced “f1” by “F”
1659: Deleted “conventional retrieval and”
1774-1778: Replaced “machine learning” by “AI”, deleted “Gaofen-3”,“(QPS)”, “(FI)”, “(BI)” and “OW”
1780-1782: Replaced “BI” by “brash ice”, “OW” by “open water”, “FI” by “floe ice”, “FI”
1786-1787: Deleted “(NI)”, “(tI)”, “(TI)”, and “(OI)”
1792-1793: Deleted “(SIC)”, “(FCN)” and “(SI)”
1796: Replace “ocean surface and SIC” by “sea ice concentration”
1804: Replace “Ocean surface and internal waves” by “ocean surface waves, internal waves”
Reviewer 4 Report
This paper mainly focusses on the summary of imaging processes and analyses of oceanic data using SAR, InSAR, PolSAR data and AI. The selected oceanic phenomena such as ocean waves, internal waves, oil slicks, currents, bathymetry, ship detection and classification, wind, aquaculture, and sea ice are detailed described.
The manuscript shows high quality and significant for understanding. The paper is strongly recommended for publication, which would be a good review reference for researcher.
Author Response
Reply to the Reviewer 4
We thank you for reviewing the manuscript.
We made corrections mainly for avoiding the repetition of same expressions already previously given.
15: “naritime” to “maritime”
89: “artificial intelligence (AI)” to “(AI)”
101: Deleted “and snow”
274: Deleted “Deep Neural Network”
282: Deleted “Convolution Neural Network”
512: Added the reference [24]
827-828: Deleted “(Automatic Identification)”
885-888: Deleted “artificial intelligence”, “support vector machine”, “neural network” and “convolution neural. network”
893-897: Added the references
960: Added the reference [234]
1483: “several” to “many”
1569-1570: Replaced “machine learning” by “AI”. Deleted “such as pattern recognition, medical diagnosis, search
engines and robotics”
1603: Replaced “f1” by “F”
1659: Deleted “conventional retrieval and”
1774-1778: Replaced “machine learning” by “AI”, deleted “Gaofen-3”,“(QPS)”, “(FI)”, “(BI)” and “OW”
1780-1782: Replaced “BI” by “brash ice”, “OW” by “open water”, “FI” by “floe ice”, “FI”
1786-1787: Deleted “(NI)”, “(tI)”, “(TI)”, and “(OI)”
1792-1793: Deleted “(SIC)”, “(FCN)” and “(SI)”
1796: Replace “ocean surface and SIC” by “sea ice concentration”
1804: Replace “Ocean surface and internal waves” by “ocean surface waves, internal waves”